# A Diffusion Theory For Deep Learning Dynamics: Stochastic Gradient Descent Exponentially Favors Flat Minima

Zeke Xie[1,2], Issei Sato [1,2], and Masashi Sugiyama[2,1]

[1]The University of Tokyo
[2]RIKEN Center for AIP
*xie@ms.k.u-tokyo.ac.jp*
*{sato,sugi}@k.u-tokyo.ac.jp*

## Abstract

Stochastic Gradient Descent (SGD) and its variants are mainstream methods for training deep networks in practice. SGD is known to find a flat minimum that often generalizes well. However, it is mathematically unclear how deep learning can select a flat minimum among so many minima. To answer the question quantitatively, we develop a density diffusion theory to reveal how minima selection quantitatively depends on the minima sharpness and the hyperparameters. To the best of our knowledge, we are the first to theoretically and empirically prove that, benefited from the Hessian-dependent covariance of stochastic gradient noise, SGD favors flat minima exponentially more than sharp minima, while Gradient Descent (GD) with injected white noise favors flat minima only polynomially more than sharp minima. We also reveal that either a small learning rate or large-batch training requires exponentially many iterations to escape from minima in terms of the ratio of the batch size and learning rate. Thus, large-batch training cannot search flat minima efficiently in a realistic computational time.

## 1 Introduction

In recent years, deep learning (LeCun et al., 2015) has achieved great empirical success in various application areas. Due to the over-parametrization and the highly complex loss landscape of deep networks, optimizing deep networks is a difficult task. Stochastic Gradient Descent (SGD) and its variants are mainstream methods for training deep networks. Empirically, SGD can usually find flat minima among a large number of sharp minima and local minima (Hochreiter & Schmidhuber, 1995; 1997). More papers reported that learning flat minima closely relate to generalization (Hardt et al., 2016; Zhang et al., 2017a; Arpit et al., 2017; Hoffer et al., 2017; Dinh et al., 2017; Neyshabur et al., 2017; Wu et al., 2017; Dziugaite & Roy, 2017; Kleinberg et al., 2018). Some researchers specifically study flatness itself. They try to measure flatness (Hochreiter & Schmidhuber, 1997; Keskar et al., 2017; Sagun et al., 2017; Yao et al., 2018), rescale flatness (Tsuzuku et al., 2019; Xie et al., 2020b), and find flatter minima (Hoffer et al., 2017; Chaudhari et al., 2017; He et al., 2019b; Xie et al., 2020a). However, we still lack a quantitative theory that answers why deep learning dynamics selects a flat minimum.

The diffusion theory is an important theoretical tool to understand how deep learning dynamics works. It helps us model the diffusion process of probability densities of parameters instead of model parameters themselves. The density diffusion process of Stochastic Gradient Langevin Dynamics (SGLD) under injected isotropic noise has been discussed by (Sato & Nakagawa, 2014; Raginsky et al., 2017; Zhang et al., 2017b; Xu et al., 2018). Zhu et al. (2019) revealed that anisotropic diffusion of SGD often leads to flatter minima than isotropic diffusion. A few papers has quantitatively studied the diffusion process of SGD under the isotropic gradient noise assumption. Jastrzębski et al. (2017) first studied the minima selection probability of SGD. Smith & Le (2018) presented a Beyesian perspective on generalization of SGD. Wu et al. (2018) studied the escape problems of

SGD from a dynamical perspective, and obtained the qualitative conclusion on the effects of batch size, learning rate, and sharpness. Hu et al. (2019) quantitatively showed that the mean escape time of SGD exponentially depends on the inverse learning rate. Achille & Soatto (2019) also obtained a related proposition that describes the mean escape time in terms of a free energy that depends on the Fisher Information. Li et al. (2017) analyzed Stochastic Differential Equation (SDE) of adaptive gradient methods. Nguyen et al. (2019) mainly contributed to closing the theoretical gap between continuous-time dynamics and discrete-time dynamics under isotropic heavy-tailed noise.

However, the related papers mainly analyzed the diffusion process under parameter-independent and isotropic gradient noise, while stochastic gradient noise (SGN) is highly parameter-dependent and anisotropic in deep learning dynamics. Thus, they failed to quantitatively formulate how SGD selects flat minima, which closely depends on the Hessian-dependent structure of SGN. We try to bridge the gap between the qualitative knowledge and the quantitative theory for SGD in the presence of parameter-dependent and anisotropic SGN. Mainly based on Theorem 3.2 , we have four contributions:

- The proposed theory formulates the fundamental roles of gradient noise, batch size, the learning rate, and the Hessian in minima selection.
- The SGN covariance is approximately proportional to the Hessian and inverse to batch size.
- Either a small learning rate or large-batch training requires exponentially many iterations to escape minima in terms of ratio of batch size and learning rate.
- To the best of our knowledge, we are the first to theoretically and empirically reveal that SGD favors flat minima exponentially more than sharp minima.

## 2 STOCHASTIC GRADIENT NOISE AND SGD DYNAMICS

We mainly introduce the necessary foundation for the proposed diffusion theory in this section. We denote the data samples as $\{x_j\}_{j=1}^{m}$, the model parameters as $\theta$ and the loss function over data samples $x$ as $L(\theta, x)$. For simplicity, we denote the training loss as $L(\theta)$. Following Mandt et al. (2017), we may write SGD dynamics as

$$\theta_{t+1} = \theta_t - \eta \frac{\partial \hat{L}(\theta_t)}{\partial \theta_t} = \theta_t - \eta \frac{\partial L(\theta_t)}{\partial \theta_t} + \eta C(\theta_t)^{\frac{1}{2}} \zeta_t, \tag{1}$$

where $\hat{L}(\theta)$ is the loss of one minibatch, $\zeta_t \sim \mathcal{N}(0, I)$, and $C(\theta)$ represents the gradient noise covariance matrix. The classic approach is to model SGN by Gaussian noise, $\mathcal{N}(0, C(\theta))$ (Mandt et al., 2017; Smith & Le, 2018; Chaudhari & Soatto, 2018).

**Stochastic Gradient Noise Analysis.** We first note that the SGN we study is introduced by minibatch training, $C(\theta_t)^{\frac{1}{2}} \zeta_t = \frac{\partial L(\theta_t)}{\partial \theta_t} - \frac{\partial \hat{L}(\theta_t)}{\partial \theta_t}$, which is the difference between gradient descent and stochastic gradient descent. According to Generalized Central Limit Theorem (Gnedenko et al., 1954), the mean of many infinite-variance random variables converges to a stable distribution, while the mean of many finite-variance random variables converges to a Gaussian distribution. As SGN is finite in practice, we believe the Gaussian approximation of SGN is reasonable.

Simsekli et al. (2019) argued that SGN is Lévy noise (stable variables), rather than Gaussian noise. They presented empirical evidence showing that SGN seems heavy-tailed, and the heavy-tailed distribution looks closer to a stable distribution than a Gaussian distribution. However, this research line (Simsekli et al., 2019; Nguyen et al., 2019) relies on a hidden strict assumption that SGN must be isotropic and obey the same distribution across dimensions. Simsekli et al. (2019) computed "SGN" across $n$ model parameters and regarded "SGN" as $n$ samples drawn from a *single-variant* distribution. This is why *one tail-index for all parameters* was studied in Simsekli et al. (2019). The arguments in Simsekli et al. (2019) did not necessarily hold for *parameter-dependent and anisotropic* Gaussian noise. In our paper, SGN computed over different minibatches obeys a *n-variant* Gaussian distribution, which can be *parameter-dependent and anisotropic*.

In Figure 1, we empirically verify that SGN is highly similar to Gaussian noise instead of heavy-tailed Lévy noise. We recover the experiment of Simsekli et al. (2019) to show that gradient noise is approximately Lévy noise only if it is computed across parameters. Figure 1 actually suggests that

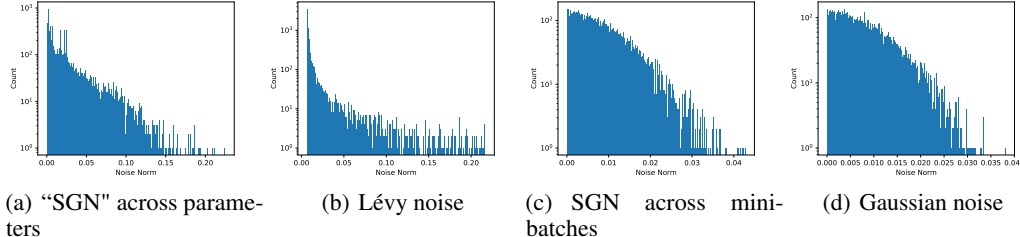

(a) "SGN" across parameters

(b) Lévy noise

(c) SGN across minibatches

(d) Gaussian noise

Figure 1: The Stochastic Gradient Noise Analysis. The histogram of the norm of the gradient noises computed with the three-layer fully-connected network on MNIST (LeCun, 1998). (a) and (c): the histograms of the norms of two kinds of gradient noise: (a) "SGN" is computed over parameters, which is actually stochastic gradient rather than SGN; (c) SGN is computed over minibatches. (b) and (d): the histograms of the norms of (scaled) Gaussian noise and Lévy noise. Based on (a) and (b), Simsekli et al. (2019) argued that gradient noise across parameters is heavy-tailed Lévy noise. Based on (c) and (d), we show that SGN without the isotropic restriction is approximately Gaussian.

the contradicted observations are from the different formulations of gradient noise. Simsekli et al. (2019) studied the distribution of SGN as a single-variant distribution, while we relax it as a $n$-variant distribution. Our empirical analysis in Figure 1 holds well at least when the batch size $B$ is larger than 16, which is common in practice. Similar empirical evidence can be observed for training ResNet18 (He et al., 2016) on CIFAR-10 (Krizhevsky et al., 2009), seen in Appendix C.

Panigrahi et al. (2019) also observed that for batch sizes 256 and above, the distribution of SGN is best described as Gaussian at-least in the early phases of training. Comparing our results with Panigrahi et al. (2019), we noticed that the Gaussianity of SGN may depend on more unknown factors. First, SGN on random models is more Gaussian than well-trained models. Second, the layer/network matters. Because SGN on some layers/networks is more Gaussian than other layers/networks.

The isotropic gradient noise assumption is too rough to capture the Hessian-dependent covariance structure of SGN, which we will study in Figure 2 later. Our theory that focuses on parameter-dependent and anisotropic SGN brings a large improvement over existing parameter-independent and isotropic noise, although Simsekli et al. (2019) brought an improvement over more conventional parameter-independent and isotropic Gaussian noise. A more sophisticated theory is interesting under parameter-independent anisotropic heavy-tailed noise, when the batch size is too small ($B \sim 1$) to apply Central Limit Theorem. We will leave it as future work.

**SGD Dynamics.** Let us replace $\eta$ by $dt$ as unit time. Then the continuous-time dynamics of SGD (Coffey & Kalmykov, 2012) is written as

$$d\theta = -\frac{\partial L(\theta)}{\partial \theta} dt + [2D(\theta)]^{\frac{1}{2}} dW_t, \tag{2}$$

where $dW_t \sim \mathcal{N}(0, I dt)$ and $D(\theta) = \frac{\eta}{2} C(\theta)$. We note that the dynamical time $t$ in the continuous-time dynamics is equal to the product of the number of iterations $T$ and the learning rate $\eta$: $t = \eta T$. The associated Fokker-Planck Equation is written as

$$\frac{\partial P(\theta, t)}{\partial t} = \nabla \cdot [P(\theta, t) \nabla L(\theta)] + \nabla \cdot \nabla D(\theta) P(\theta, t) \tag{3}$$

$$= \sum_i \frac{\partial}{\partial \theta_i} \left[ P(\theta, t) \frac{\partial L(\theta)}{\partial \theta_i} \right] + \sum_i \sum_j \frac{\partial^2}{\partial \theta_i \partial \theta_j} D_{ij}(\theta) P(\theta, t), \tag{4}$$

where $\nabla$ is a nabla operator, and $D_{ij}$ is the element in the $i$th row and $j$th column of $D$. In standard SGLD, the injected gradient noise is fixed and isotropic Gaussian, $D = I$.

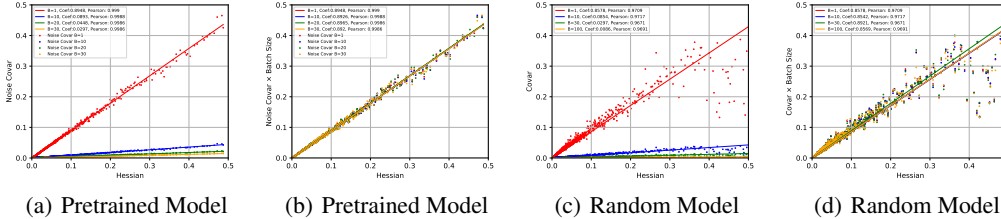

|     |     |     |     |
| --- | --- | --- | --- |
| (a) Pretrained Model | (b) Pretrained Model | (c) Random Model | (d) Random Model |

Figure 2: We empirically verified $C(\theta) = \frac{H(\theta)}{B}$ by using three-layer fully-connected network on MNIST (LeCun, 1998). The pretrained Models are usually near critical points, while randomly Initialized Models are far from critical points. We display all elements $H_{(i,j)} \in [1e-4, 0.5]$ of the Hessian matrix and the corresponding elements $C_{(i,j)}$ of gradient noise covariance matrix in the space spanned by the eigenvectors of Hessian. Another supplementary experiment on Avila Dataset (De Stefano et al., 2018) in Appendix C reports $\hat{C}_{avila} \approx 1.004 \frac{H}{B}$. The small difference factor between the empirical result and the ideal Equation is mainly because the pretrained network is not perfectly located at a critical point.

The next question is how to formulate the SGN covariance $C(\theta)$ for SGD? Based on Smith & Le (2018), we can express the SGN covariance as

$$C(\theta) = \frac{1}{B}\left[\frac{1}{m}\sum_{j=1}^{m}\nabla L(\theta, x_j)\nabla L(\theta, x_j)^\top - \nabla L(\theta)\nabla L(\theta)^\top\right] \approx \frac{1}{Bm}\sum_{j=1}^{m}\nabla L(\theta, x_j)\nabla L(\theta, x_j)^\top.$$

(5)

The approximation is true near critical points, due to the fact that the gradient noise variance dominates the gradient mean near critical points. We know the observed fisher information matrix satisfies $\mathrm{FIM}(\theta) \approx H(\theta)$ near minima, referring to Chapter 8 of (Pawitan, 2001). Following Jastrzębski et al. (2017); Zhu et al. (2019), we obtain

$$C(\theta) \approx \frac{1}{Bm}\sum_{j=1}^{m}\nabla L(\theta, x_j)\nabla L(\theta, x_j)^\top = \frac{1}{B}\mathrm{FIM}(\theta) \approx \frac{1}{B}H(\theta),$$

(6)

which approximately gives

$$D(\theta) = \frac{\eta}{2}C(\theta) = \frac{\eta}{2B}H(\theta)$$

(7)

near minima. It indicates that the SGN covariance $C(\theta)$ is approximately proportional to the Hessian $H(\theta)$ and inverse to the batch size $B$. Obviously, we can generalize Equation 7 by $D(\theta) = \frac{\eta C(\theta)}{2} = \frac{\eta}{2B}[H(\theta)]^+$ near critical points, when there exist negative eigenvalues in $H$ along some directions. We use $[\cdot]^+$ to denote the positive semidefinite transformation of a symmetric matrix: if we have the eigendecomposition $H = U\,\mathrm{diag}(H_1, \cdots, H_{n-1}, H_n)U^\top$, then $[H]^+ = U\,\mathrm{diag}(|H_1|, \cdots, |H_{n-1}|, |H_n|)U^\top$.

We empirically verify this relation in Figure 2 for pretrained fully-connected networks, and a follow-up paper Xie et al. (2020c) first verified this relation for randomly initialized fully-connected networks on real-world datasets. The Pearson Correlation is up to 0.999 for pretrained networks. We note that, the relation still approximately holds for even the randomly network, which is far from critical points. The correlation is especially high along the flat directions with small-magnitude eigenvalues of the Hessian (Xie et al., 2020c). We emphasize that previous papers with the isotropic Lévy or Gaussian noise approximation all failed to capture this core relation in deep learning dynamics.

## 3 SGD DIFFUSION THEORY

We start the theoretical analysis from the classical Kramers Escape Problem (Kramers, 1940). We assume there are two valleys, Sharp Valley $a_1$ and Flat Valley $a_2$, seen in Figure 3. Also Col b is the

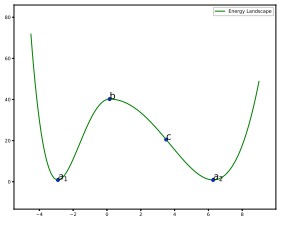 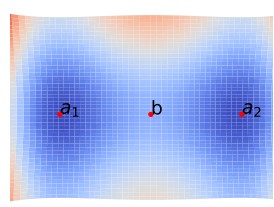

|                       |                        |
| :-------------------: | :--------------------: |
| (a) 1-Dimensional Escape | (b) High-Dimensional Escape |

Figure 3: Kramers Escape Problem. $a_1$ and $a_a$ are minima of two neighboring valleys. $b$ is the saddle point separating the two valleys. $c$ locates outside of Valley $a_1$.

boundary between two valleys. What is the mean escape time for a particle governed by Equation 2 to escape from Sharp Valley $a_1$ to Flat Valley $a_2$? The mean escape time is widely used in related statistical physics and stochastic process (Van Kampen, 1992; Nguyen et al., 2019).

Gauss's Divergence Theorem (Arfken & Weber, 1999; Lipschutz et al., 2009) states that the surface integral of a vector field over a closed surface, which is called the flux through the surface, is equal to the volume integral of the divergence over the region inside the surface. We respectively denote the mean escape time as $\tau$, the escape rate as $\gamma$, and the probability current as $J$. We apply Gauss's Divergence Theorem to the Fokker-Planck Equation resulting in

$$\nabla \cdot [P(\theta,t)\nabla L(\theta)] + \nabla \cdot \nabla D(\theta)P(\theta,t) = \frac{\partial P(\theta,t)}{\partial t} = -\nabla \cdot J(\theta,t). \tag{8}$$

The mean escape time is expressed (Van Kampen, 1992) as

$$\tau = \frac{1}{\gamma} = \frac{P(\theta \in V_a)}{\int_{S_a} J \cdot dS}, \tag{9}$$

where $P(\theta \in V_a) = \int_{V_a} P(\theta)dV$ is the current probability inside Valley a, $J$ is the probability current produced by the probability source $P(\theta \in V_a)$, $j = \int_{S_a} J \cdot dS$ is the probability flux (surface integrals of probability current), $S_a$ is the surface (boundary) surrounding Valley a, and $V_a$ is the volume surrounded by $S_a$. We have $j = J$ in the case of one-dimensional escape.

**Classical Assumptions.** We state three classical assumptions first for the density diffusion theory. Assumption 1 is the common second order Taylor approximation, which was also used by (Mandt et al., 2017; Zhang et al., 2019). Assumptions 2 and 3 are widely used in many fields' Kramers Escape Problems, including statistical physics (Kramers, 1940; Hanggi, 1986), chemistry (Eyring, 1935; Hänggi et al., 1990), biology (Zhou, 2010), electrical engineering (Coffey & Kalmykov, 2012), and stochastic process (Van Kampen, 1992; Berglund, 2013). Related machine learning papers (Jastrzębski et al., 2017) usually used Assumptions 2 and 3 as the background of Kramers Escape Problems.

**Assumption 1** (The Second Order Taylor Approximation). *The loss function around critical points $\theta^\star$ can be approximately written as*

$$L(\theta) = L(\theta^\star) + g(\theta^\star)(\theta - \theta^\star) + \frac{1}{2}(\theta - \theta^\star)^\top H(\theta^\star)(\theta - \theta^\star).$$

**Assumption 2** (Quasi-Equilibrium Approximation). *The system is in quasi-equilibrium near minima.*

**Assumption 3** (Low Temperature Approximation). *The gradient noise is small (low temperature).*

We will dive into these two assumptions deeper than previous papers for SGD dynamics. Assumptions 2 and 3 both mean that our diffusion theory can better describe the escape processes that cost more iterations. As this class of "slow" escape processes takes main computational time compared with "fast" escape processes, this class of "slow" escape process is more interesting for training of deep neural networks. Our empirical analysis in Section 4 supports that the escape processes in the

wide range of iterations (50 to 100,000 iterations) can be modeled by our theory very well. Thus, Assumption 2 and 3 are reasonable in practice. More discussion can be found in Appendix B.

**Escape paths.** We generalize the concept of critical points into critical paths as the path where 1) the gradient perpendicular to the path direction must be zero, and 2) the second order directional derivatives perpendicular to the path direction must be nonnegative. The Most Possible Paths (MPPs) for escaping must be critical paths. The most possible escape direction at one point must be the direction of one eigenvector of the Hessian at the point. Under Assumption 3, the probability density far from critical points and MPPs is very small. Thus, the density diffusion will concentrate around MPPs. Draxler et al. (2018) reported that minima in the loss landscape of deep networks are connected by Minimum Energy Paths (MEPs) that are essentially flat and Local MEPs that have high-loss saddle points. Obviously, MPPs in our paper correspond to Local MEPs. The density diffusion along MEPs, which are strictly flat, is ignorable according to our following analysis.

The boundary between Sharp Valley $a_1$ and Flat Valley $a_2$ is the saddle point $b$. The Hessian at $b$, $H_b$, must have only one negative eigenvalue and the corresponding eigenvector is the escape direction. Without losing generality, we first assume that there is only one most possible path through Col $b$ existing between Sharp Valley $a_1$ and Flat Valley $a_2$.

**SGLD diffusion.** We first analyze a simple case: how does SGLD escape sharp minima? Researchers are interested in SGLD, when the injected noise dominates SGN as $\eta \to 0$ in final epochs. Because SGLD may work as a Bayesian inference method in this limit (Welling & Teh, 2011). SGLD is usually simplified as Gradient Descent with injected white noise, whose behavior is identical to Kramers Escape Problem with thermo noise in statistical physics. We present Theorem 3.1. We leave the proof in Appendix A.1. We also note that more precise SGLD diffusion analysis should study a mixture of injected white noise and SGN.

**Theorem 3.1** (SGLD Escapes Minima). *The loss function $L(\theta)$ is of class $C^2$ and n-dimensional. Only one most possible path exists between Valley a and the outside of Valley a. If Assumption 1, 2, and 3 hold, and the dynamics is governed by SGLD, then the mean escape time from Valley a to the outside of Valley a is*

$$\tau = \frac{1}{\gamma} = 2\pi \sqrt{\frac{-\det(H_b)}{\det(H_a)}} \frac{1}{|H_{be}|} \exp\left(\frac{\Delta L}{D}\right).$$

*We denote that $H_a$ and $H_b$ are the Hessians of the loss function at the minimum $a$ and the saddle point $b$, $\Delta L = L(b) - L(a)$ is the loss barrier height, $e$ indicates the escape direction, and $H_{be}$ is the eigenvalue of the Hessian $H_b$ corresponding to the escape direction. The diffusion coefficient $D$ is usually set to $1$ in SGLD.*

**SGD diffusion.** However, SGD diffusion is essentially different from SGLD diffusion in several aspects: 1) anisotropic noise, 2) parameter-dependent noise, and 3) the stationary distribution of SGD is far from the Gibs-Boltzmann distribution, $P(\theta) = \frac{1}{Z} \exp\left(-\frac{L(\theta)}{D}\right)$. These different characteristics make SGD diffusion behave differently from known physical dynamical systems and much less studied than SGLD diffusion. We formulate Theorem 3.2 for SGD. We leave the proof in Appendix A.2.The theoretical analysis of SGD can be easily generalized to the dynamics with a mixture of SGN and injected white noise, as long as the eigenvectors of $D(\theta)$ are closely aligned with the eigenvectors of $H(\theta)$.

**Theorem 3.2** (SGD Escapes Minima). *The loss function $L(\theta)$ is of class $C^2$ and n-dimensional. Only one most possible path exists between Valley a and the outside of Valley a. If Assumption 1, 2, and 3 hold, and the dynamics is governed by SGD, then the mean escape time from Valley a to the outside of Valley a is*

$$\tau = 2\pi \frac{1}{|H_{be}|} \exp\left[\frac{2B\Delta L}{\eta}\left(\frac{s}{H_{ae}} + \frac{(1-s)}{|H_{be}|}\right)\right],$$

*where $s \in (0, 1)$ is a path-dependent parameter, and $H_{ae}$ and $H_{be}$ are, respectively, the eigenvalues of the Hessians at the minimum $a$ and the saddle point $b$ corresponding to the escape direction $e$.*

**Multiple-path escape.** Each escape path contributes to the total escape rate. Multiple paths combined together have a total escape rate. If there are multiple parallel from the start valley to the end valley, we can compute the total escape rate easily based on the following computation rule. The computation

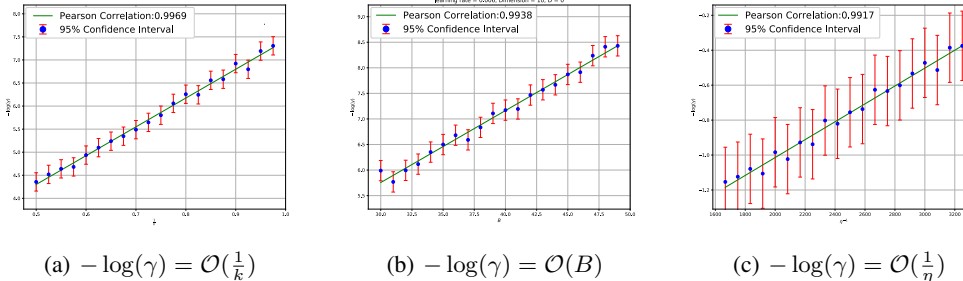

(a) $-\log(\gamma) = \mathcal{O}(\frac{1}{k})$     (b) $-\log(\gamma) = \mathcal{O}(B)$     (c) $-\log(\gamma) = \mathcal{O}(\frac{1}{\eta})$

Figure 4: The mean escape time analysis of SGD by using Styblinski-Tang Function. The Pearson Correlation is higher than $0.99$. Left Column: Sharpness. Middle Column: Batch Size. Right Column: Learning Rate.

rule is based on the fact that the probability flux integrals are additive. We can easily generalize the mean escape time analysis into the cases that there are multiple parallel escape paths indexed by $p$. As for multiple-valley escape problems, we can always reduce a multiple-valley escape problem into multiple two-valley escape problems. We also note that, while Theorem A.2 does not depend the dimensionality directly, higher dimensionality may increase the number of escape paths and loss valleys, and change the spectrum of the Hessians.

**Rule 1.** *If there are multiple MPPs between the start valley and the end valley, then $\gamma_{total} = \sum_p \gamma_p$.*

Thus, we only need to find the saddle points that connect two valleys as we analyzed in the paper and analyze the escape rates.

**Minima selection.** Now, we may formulate the probability of minima selection as Proposition 1. We leave the proof in Appendix A.3. In deep learning, one loss valley represents one mode and the landscape contain many good modes and bad modes. SGD transits from one mode to another mode during training. The mean escape time of one mode corresponds to the number of iterations which SGD spends on this mode during training, which is naturally proportional to the probability of selecting this mode after training.

**Proposition 1.** *Suppose there are two valleys connected by an escape path. If all assumptions of Theorem 3.2 hold, then the stationary distribution of locating these valleys is given by*

$$P(\theta \in V_a) = \frac{\tau_a}{\sum_v \tau_v},$$

*where $v$ is the index of valleys, and $\tau_v$ is the mean escape time from Valley $v$ to the outside of Valley $v$.*

## 4   EMPIRICAL ANALYSIS

In this section, we try to directly validate the escape formulas on real-world datasets. Each escape process, from the inside of loss valleys to the outside of loss valleys, are repeatedly simulated for 100 times under various gradient noise scales, batch sizes, learning rates, and sharpness.

How to compare the escape rates under the same settings with various minima sharpness? Our method is to multiply a rescaling factor $\sqrt{k}$ to each parameter, and the Hessian will be proportionally rescaled by a factor $k$. If we let $L(\theta) = f(\theta) \rightarrow L(\theta) = f(\sqrt{k}\theta)$, then $H(\theta) = \nabla^2 f(\theta) \rightarrow H(\theta) = k\nabla^2 f(\theta)$. Thus, we can use $k$ to indicate the minima sharpness. The theoretical relations of SGD we try to validate can be formulated as: (1) $-\log(\gamma) = \mathcal{O}(\frac{1}{k})$, (2) $-\log(\gamma) = \mathcal{O}(B)$, and (3) $-\log(\gamma) = \mathcal{O}(\frac{1}{\eta})$.

**The mean escape time analysis of SGD.** Styblinski-Tang Function, which has multiple minima and saddle points, is a common test function for nonconvex optimization. We conduct an intuitional 10-dimensional experiment, where the simulations start from a given minimum and terminate when reaching the boundary of the loss valley. The number of iterations is recorded for calculating

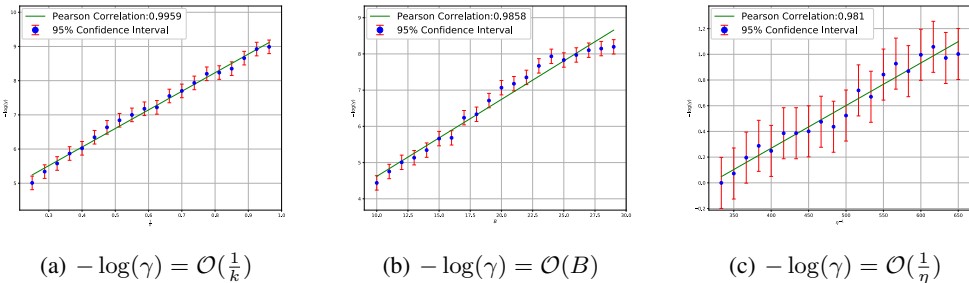

(a) $-\log(\gamma) = \mathcal{O}(\frac{1}{k})$     (b) $-\log(\gamma) = \mathcal{O}(B)$     (c) $-\log(\gamma) = \mathcal{O}(\frac{1}{\eta})$

Figure 5: The mean escape time analysis of SGD by training neural networks on Avila Dataset. Left Column: Sharpness. Middle Column:Batch Size. Right Column: Learning Rate. We leave the results on Banknote Authentication, Cardiotocography, and Sensorless Drive Diagnosis in Appendix D.

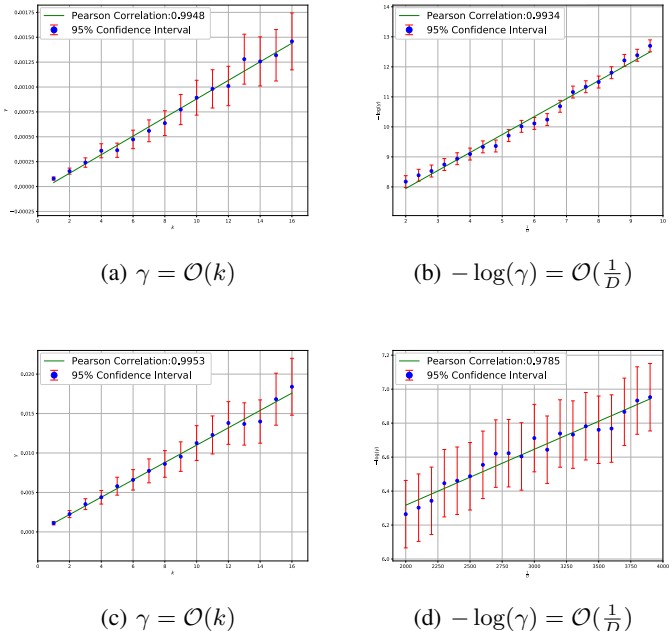

(a) $\gamma = \mathcal{O}(k)$     (b) $-\log(\gamma) = \mathcal{O}(\frac{1}{D})$

(c) $\gamma = \mathcal{O}(k)$     (d) $-\log(\gamma) = \mathcal{O}(\frac{1}{D})$

Figure 6: The mean escape time analysis of SGLD. Subfigure (a) and (b): Styblinski-Tang Function. Subfigure (c) and (d): Neural Network.

the escape rate. We also train fully connected networks on four real-world datasets, including a) Avila, b) Banknote Authentication, c) Cardiotocography, d) Dataset for Sensorless Drive Diagnosis (De Stefano et al., 2018; Dua & Graff, 2017). Figure 4 and Figure 5 clearly verifies that the escape rate exponentially depends on the minima sharpness (reflected by $k$), the batch size, and the learning rate on both test functions and real-world training, which fully supports our theoretical results.

Model architecture and details: We used fully-connected networks with the depth 2 and the width 10 in Figure 5. The experiments using Logistic Regression and Fully-connected networks with the depth 3 are presented in Appendix E. We leave more experimental details and results in Appendix D.1 and Appendix E.

**The mean escape time analysis of SGLD.** We try to validate $\gamma = \mathcal{O}(k)$ and $-\log(\gamma) = \mathcal{O}(\frac{1}{D})$ for SGLD (dominated by injected Gaussian noise). Figure 6 shows that SGLD only favors flat minima polynomially more than sharp minima as Theorem 3.1 indicates. Figure 6 also verifies that the injected gradient noise scale exponentially affects flat minima selection.

## 5    DISCUSSION

**SGD favors flat minima exponentially more than sharp minima.** We can discover a few interesting insights about SGD by Theorem 3.2. Most importantly, the mean escape time exponentially depends on the eigenvalue of the Hessian at minima along the escape direction, $H_{ae}$. Thus, SGD favors flat minima exponentially more than sharp minima. We claim one main advantage of SGD comes from the exponential relation of the mean escape time and the minima sharpness. The measure of "sharpness" has reformed in contexts of SGLD and SGD. In the context of SGLD, the "sharpness" is quantified by the determinant of the Hessian. In the context of SGD, the "sharpness" is quantified by the top eigenvalues of the Hessian along the escape direction. Based on the proposed diffusion theory, recent work (Xie et al., 2020c) successfully proved that SGD favors flat minima significantly more than Adam.

**The ratio of the batch size and the learning rate exponentially matters.** Theorem 3.2 explains why large-batch training can easily get trapped near sharp minima, and increasing the learning rate proportionally is helpful for large-batch training (Krizhevsky, 2014; Keskar et al., 2017; Sagun et al., 2017; Smith et al., 2018; Yao et al., 2018; He et al., 2019a). We argue that the main cause is large-batch training expects exponentially longer time to escape minima. Note that, as the mean escape time in the theorems is equivalent to the product of the learning rate and the number of iterations, both the number of iterations and dynamical time exponentially depend on the ratio of the batch size and the learning rate. The practical computational time in large-batch training is usually too short to search many enough flat minima. We conjecture that exponentially increasing training iterations may be helpful for large batch training, while this is often too expensive in practice.

**Low dimensional diffusion.** Most eigenvalues of the Hessian at the loss landscape of overparametrized deep networks are close to zero, while only a small number of eigenvalues are large (Sagun et al., 2017; Li et al., 2018). Zero eigenvalues indicate zero diffusion along the corresponding directions. Thus, we may theoretically ignore these zero-eigenvalue directions. This also indicates that the density diffusion is ignorable along an essentially flat MEP in Draxler et al. (2018).

As the escape rate exponentially depends the corresponding eigenvalues, a small number of large eigenvalues means that the process of minima selection mainly happens in the relatively low dimensional subspace corresponding to top eigenvalues of the Hessian. Gur-Ari et al. (2018) also reported a similar finding. Although the parameter space is very high-dimensional, SGD dynamics hardly depends on those "meaningless" dimensions with small second order directional derivatives. This novel characteristic of SGD significantly reduces the explorable parameter space around one minimum into a much lower dimensional space.

**High-order effects.** As we have applied the second-order Taylor approximation near critical points, our SGD diffusion theory actually excludes the third-order and higher-order effect. The asymmetric valley in He et al. (2019b), which only appears in high-order analysis, is beyond the scope of this paper. However, we also argue that the third-order effect is much smaller than the second-order effect under the low temperature assumption in Kramers Escape Problems. We will leave the more refined high-order theory as future work.

## 6    CONCLUSION

In this paper, we demonstrate that one essential advantage of SGD is selecting flat minima with an exponentially higher probability than sharp minima. To the best of our knowledge, we are the first to formulate the exponential relation of minima selection to the minima sharpness, the batch size, and the learning rate. Our work bridges the gap between the qualitative knowledge and the quantitative theoretical knowledge on the minima selection mechanism of SGD. We believe the proposed theory not only helps us understand how SGD selects flat minima, but also will provide researchers a powerful theoretical tool to analyze more learning behaviors and design better optimizers in future.

### ACKNOWLEDGEMENT

We thanks Dr. Yuanqian Tang for helpful discussion. MS was supported by the International Research Center for Neurointelligence (WPI-IRCN) at The University of Tokyo Institutes for Advanced Study.

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

## A  PROOFS

### A.1  PROOF OF THEOREM 3.1

*Proof.* This proposition is a well known conclusion in statistical physics under Assumption 1, 2 and 3. We still provide an intuitional proof here, and the following proof of SGD Diffusion will closely relate to this proof. We decompose the proof into two steps: 1) compute the probability of locating in valley a, $P(\theta \in V_a)$, and 2) compute the probability flux $j = \int_{S_a} J \cdot dS$.

Without losing generality, we first prove the one-dimensional case.

Step 1: Under Assumption 1, the stationary distribution around minimum a is $P(\theta) = P(a) \exp[-\frac{L(\theta)-L(a)}{T}]$, where $T = D$. Under Assumption 3, we may only consider the second order Taylor approximation of the density function around critical points. We use the $T$ notation as the temperature parameter in the stationary distribution, and use the $D$ notation as the diffusion coefficient in the dynamics, for their different roles.

$$P(\theta \in V_a) \tag{10}$$

$$= \int_{\theta \in V_a} P(\theta) dV \tag{11}$$

$$= \int_{\theta \in V_a} P(a) \exp\left[-\frac{L(\theta)-L(a)}{T}\right] d\theta \tag{12}$$

$$= P(a) \int_{\theta \in V_a} \exp\left[-\frac{\frac{1}{2}(\theta-a)^\top H_a(\theta-a) + \mathcal{O}(\Delta\theta^3)}{T}\right] d\theta \tag{13}$$

$$= P(a) \frac{(2\pi T)^{\frac{1}{2}}}{H_a^{\frac{1}{2}}}. \tag{14}$$

Step 2:

$$J = P(\theta)\nabla L(\theta) + P(\theta)\nabla D + D\nabla P(\theta) \tag{15}$$

$$J = P(\theta)\left(\nabla L(\theta) + \nabla D - \frac{D}{T}\nabla L(\theta)\right) \tag{16}$$

$$\nabla D = \left(\frac{D}{T} - 1\right)\nabla L \tag{17}$$

Apply this result to the Fokker-Planck Equation 4, we have

$$\nabla \cdot \nabla[D(\theta)P(\theta,t)] \tag{18}$$

$$= \nabla \cdot D\nabla P(\theta,t) + \nabla \cdot \left[\left(\frac{D}{T}-1\right)\nabla L(\theta)\right]P(\theta,t) \tag{19}$$

And thus we obtain the Smoluchowski equation and a new form of J

$$\frac{\partial P(\theta, t)}{\partial t} = \nabla \cdot \left[ D \left( \frac{1}{T} \nabla L(\theta) + \nabla \right) P(\theta, t) \right] = -\nabla \cdot J(\theta, t), \tag{20}$$

$$J(\theta) = D \exp \left( \frac{-L(\theta)}{T} \right) \nabla \left[ \exp \left( \frac{L(\theta)}{T} \right) P(\theta) \right]. \tag{21}$$

We note that the probability density outside Valley a must be zero, $P(c) = 0$. As we want to compute the probability flux escaping from Valley a in the proof, the probability flux escaping from other valleys into Valley a should be ignored. Under Assumption 2, we integrate the equation from Valley a to the outside of Valley a along the most possible escape path

$$\int_a^c \frac{\partial}{\partial \theta} \left[ \exp \left( \frac{L(\theta)}{T} \right) P(\theta) \right] d\theta = \int_a^c -\frac{J}{D} \exp \left( \frac{L(\theta)}{T} \right) d\theta \tag{22}$$

$$\exp \left( \frac{L(\theta)}{T} \right) P(\theta) \Big|_a^c = -\frac{J}{D} \int_a^c \exp \left( \frac{L(\theta)}{T} \right) d\theta \tag{23}$$

$$0 - \exp \left( \frac{L(a)}{T} \right) P(a) = -\frac{J}{D} \int_a^c \exp \left( \frac{L(\theta)}{T} \right) d\theta \tag{24}$$

$$J = \frac{D \exp \left( \frac{L(a)}{T} \right) P(a)}{\int_a^c \exp \left( \frac{L(\theta)}{T} \right) d\theta}. \tag{25}$$

We move $J$ to the outside of integral based on Gauss's Divergence Theorem, because $J$ is fixed on the escape path from one minimum to another. As there is no field source on the escape path, $\int_V \nabla \cdot J(\theta) dV = 0$. Then $\nabla J(\theta) = 0$. Obviously, only minima are probability sources in deep learning. Under Assumption 3 and the second-order Taylor approximation, we have

$$\int_a^c \exp \left( \frac{L(\theta)}{T} \right) d\theta \tag{26}$$

$$= \int_a^c \exp \left[ \frac{L(b) + \frac{1}{2}(\theta - b)^\top H_b (\theta - b) + \mathcal{O}(\Delta \theta^3)}{T} \right] d\theta \tag{27}$$

$$\approx \exp \left( \frac{L(b)}{T} \right) \int_{-\infty}^{+\infty} \exp \left[ \frac{\frac{1}{2}(\theta - b)^\top H_b (\theta - b)}{T} \right] d\theta \tag{28}$$

$$= \exp \left( \frac{L(b)}{T} \right) \sqrt{\frac{2\pi T}{|H_b|}}. \tag{29}$$

Based on the results of Step 1 and Step 2, we obtain

$$\gamma = \frac{\int_{S_a} J \cdot dS}{P(\theta \in V_a)} = \frac{J}{P(\theta \in V_a)} \tag{30}$$

$$= \frac{D P(a) \exp \left( \frac{L(a)}{T} \right)}{\exp \left( \frac{L(b)}{T} \right) \sqrt{\frac{2\pi T}{|H_b|}}} \frac{1}{P(a) \sqrt{\frac{2\pi T}{H_a}}} \tag{31}$$

$$= \frac{D \sqrt{H_a |H_b|}}{2\pi T} \exp \left( -\frac{\Delta L_{ab}}{T} \right) \tag{32}$$

$$= \frac{\sqrt{H_a |H_b|}}{2\pi} \exp \left( -\frac{\Delta L_{ab}}{D} \right) \tag{33}$$

We generalize the proof of one-dimensional diffusion to high-dimensional diffusion

Step 1:

$$P(\theta \in V_a) \tag{34}$$

$$= \int_{\theta \in V_a} P(\theta) dV \tag{35}$$

$$= \int_{\theta \in V_a} P(a) \exp\left[-\frac{L(\theta) - L(a)}{T}\right] dV \tag{36}$$

$$= P(a) \int_{\theta \in V_a} \exp\left[-\frac{\frac{1}{2}(\theta - a)^\top H_a(\theta - a) + \mathcal{O}(\Delta\theta^3)}{T}\right] dV \tag{37}$$

$$= P(a) \frac{(2\pi T)^{\frac{n}{2}}}{\det(H_a)^{\frac{1}{2}}} \tag{38}$$

Step 2: Based on the formula of the one-dimensional probability current and flux, we obtain

$$\int_{S_b} J \cdot dS \tag{39}$$

$$= J_b \int_{S_b} \exp\left[-\frac{\frac{1}{2}(\theta - b)^\top H_b^+(\theta - b)}{T}\right] dS \tag{40}$$

$$= J_b \frac{(2\pi T)^{\frac{n-1}{2}}}{(\prod_{i=1}^{n-1} H_{bi})^{\frac{1}{2}}} \tag{41}$$

So we have

$$\tau = 2\pi \sqrt{\frac{\prod_{i=1}^{n-1} H_{bi}}{\det(H_a)|H_{be}|}} \exp\left(\frac{\Delta L}{T}\right) \tag{42}$$

$$= 2\pi \sqrt{\frac{-\det(H_b)}{\det(H_a)}} \frac{1}{|H_{be}|} \exp\left(\frac{\Delta L}{D}\right). \tag{43}$$

$\square$

## A.2  Proof of Theorem 3.2

*Proof.* We decompose the proof into two steps and analyze the one-dimensional case like before. The following proof is similar to the proof of SGLD except that we make $T_a$ the temperature near the minimum a and $T_b$ the temperature near the saddle point b.

One-dimensional SGD Diffusion:

Step 1: Under Assumption 3, we may only consider the second order Taylor approximation of the density function around critical points.

$$P(\theta \in V_a) \tag{44}$$

$$= \int_{\theta \in V_a} P(\theta) dV \tag{45}$$

$$= \int_{\theta \in V_a} P(a) \exp\left[-\frac{L(\theta) - L(a)}{T_a}\right] dV \tag{46}$$

$$= P(a) \int_{\theta \in V_a} \exp\left[-\frac{\frac{1}{2}(\theta - a)^\top H_a(\theta - a) + \mathcal{O}(\Delta\theta^3)}{T_a}\right] d\theta \tag{47}$$

$$= P(a) \frac{(2\pi T_a)^{\frac{1}{2}}}{H_a^{\frac{1}{2}}} \tag{48}$$

Step 2:

$$J = P(\theta)\nabla L(\theta) + P(\theta)\nabla D + D\nabla P(\theta) \tag{49}$$

$$J = P(\theta)\left[\nabla L(\theta) + \nabla D - \frac{D}{T}\nabla L(\theta) - DL(\theta)\nabla\left(\frac{1}{T}\right)\right] \tag{50}$$

According to Equation 7, $\nabla\left(\frac{1}{T}\right)$ is ignorable near the minimum a and the col b, thus

$$\nabla D = \left(\frac{D}{T} - 1\right)\nabla L. \tag{51}$$

Apply this result to the Fokker-Planck Equation 4, we have

$$\nabla \cdot \nabla[D(\theta)P(\theta,t)] \tag{52}$$

$$= \nabla \cdot D\nabla P(\theta,t) + \nabla \cdot \left[\left(\frac{D}{T} - 1\right)\nabla L(\theta)\right]P(\theta,t) \tag{53}$$

And thus we obtain the Smoluchowski equation and a new form of J

$$\frac{\partial P(\theta,t)}{\partial t} = \nabla \cdot \left[D\left(\frac{1}{T}\nabla L(\theta) + \nabla\right)P(\theta,t)\right] = -\nabla \cdot J, \tag{54}$$

$$J = D\exp\left(\frac{-L(\theta)}{T}\right)\nabla\left[\exp\left(\frac{L(\theta)}{T}\right)P(\theta)\right]. \tag{55}$$

We note that the Smoluchowski equation is true only near critical points. We assume the point s is the midpoint on the most possible path between a and b, where $L(s) = (1-s)L(a) + sL(b)$. The temperature $T_a$ dominates the path $a \to s$, while temperature $T_b$ dominates the path $s \to b$. So we have

$$\nabla\left[\exp\left(\frac{L(\theta) - L(s)}{T}\right)P(\theta)\right] = JD^{-1}\exp\left(\frac{L(\theta) - L(s)}{T}\right). \tag{56}$$

Under Assumption 2, we integrate the equation from Valley a to the outside of Valley a along the most possible escape path

$$Left = \int_a^c \frac{\partial}{\partial\theta}[\exp\left(\frac{L(\theta) - L(s)}{T}\right)P(\theta)]d\theta \tag{57}$$

$$= \int_a^s \frac{\partial}{\partial\theta}\left[\exp\left(\frac{L(\theta) - L(s)}{T_a}\right)P(\theta)\right]d\theta \tag{58}$$

$$+ \int_s^c \frac{\partial}{\partial\theta}\left[\exp\left(\frac{L(\theta) - L(s)}{T_b}\right)P(\theta)\right]d\theta \tag{59}$$

$$= [P(s) - \exp\left(\frac{L(a) - L(s)}{T_a}\right)P(a)] + [0 - P(s)] \tag{60}$$

$$= -\exp\left(\frac{L(a) - L(s)}{T_a}\right)P(a) \tag{61}$$

$$Right = -J\int_a^c D^{-1}\exp\left(\frac{L(\theta) - L(s)}{T}\right)d\theta \tag{62}$$

We move $J$ to the outside of integral based on Gauss's Divergence Theorem, because $J$ is fixed on the escape path from one minimum to another. As there is no field source on the escape path, $\int_V \nabla \cdot J(\theta)dV = 0$ and $\nabla J(\theta) = 0$. Obviously, only minima are probability sources in deep learning. So we obtain

$$J = \frac{\exp\left(\frac{L(a) - L(s)}{T_a}\right)P(a)}{\int_a^c D^{-1}\exp\left(\frac{L(\theta) - L(s)}{T}\right)d\theta}. \tag{63}$$

Under Assumption 3, we have

$$\int_a^c D^{-1} \exp\left(\frac{L(\theta) - L(s)}{T}\right) d\theta \tag{64}$$

$$\approx \int_a^c D^{-1} \exp\left[\frac{L(b) - L(s) + \frac{1}{2}(\theta - b)^\top H_b(\theta - b)}{T_b}\right] d\theta \tag{65}$$

$$\approx D_b^{-1} \int_{-\infty}^{+\infty} \exp\left[\frac{L(b) - L(s) + \frac{1}{2}(\theta - b)^\top H_b(\theta - b)}{T_b}\right] d\theta \tag{66}$$

$$= D_b^{-1} \exp\left(\frac{L(b) - L(s)}{T_b}\right) \sqrt{\frac{2\pi T_b}{|H_b|}}. \tag{67}$$

Based on the results of Step 1 and Step 2, we have

$$\gamma = \frac{\int_{S_a} J \cdot dS}{P(\theta \in V_a)} = \frac{J}{P(\theta \in V_a)} \tag{68}$$

$$= \frac{P(a) \exp\left(\frac{L(a) - L(s)}{T_a}\right)}{D_b^{-1} \exp\left(\frac{L(b) - L(s)}{T_b}\right) \sqrt{\frac{2\pi T_b}{|H_b|}}} \frac{1}{P(a)\sqrt{\frac{2\pi T_a}{H_a}}} \tag{69}$$

$$= \frac{\sqrt{T_b H_a |H_b|}}{2\pi \sqrt{T_a}} \exp\left(-\frac{L(s) - L(a)}{T_a} - \frac{L(b) - L(s)}{T_b}\right) \tag{70}$$

$$= \frac{\sqrt{T_b H_a |H_b|}}{2\pi \sqrt{T_a}} \exp\left(-\frac{s\Delta L}{T_a} - \frac{(1-s)\Delta L}{T_b}\right) \tag{71}$$

So we have

$$\tau = \frac{1}{\gamma} = 2\pi \sqrt{\frac{T_a}{T_b H_a |H_b|}} \exp\left(\frac{s\Delta L}{T_a} + \frac{(1-s)\Delta L}{T_b}\right). \tag{72}$$

In the case of pure SGN, $T_a = \frac{\eta}{2B} H_a$ and $T_b = -\frac{\eta}{2B} H_b$ gives

$$\tau = \frac{1}{\gamma} = 2\pi \frac{1}{|H_b|} \exp\left[\frac{2B\Delta L}{\eta}\left(\frac{s}{H_a} + \frac{(1-s)}{|H_b|}\right)\right]. \tag{73}$$

We generalize the proof above into the high-dimensional SGD diffusion.

Step 1:

$$P(\theta \in V_a) \tag{74}$$

$$= \int_{\theta \in V_a} P(\theta) dV \tag{75}$$

$$= P(a) \int_{\theta \in V_a} \exp\left[-\frac{1}{2}(\theta - a)^\top (D_a^{-\frac{1}{2}} H_a D_a^{-\frac{1}{2}})(\theta - a)\right] dV \tag{76}$$

$$= P(a) \frac{(2\pi)^{\frac{n}{2}}}{\det(D_a^{-1} H_a)^{\frac{1}{2}}} \tag{77}$$

Step 2: Based on the formula of the one-dimensional probability current and flux, we obtain the high-dimensional flux escaping through Col b:

$$\int_{S_b} J \cdot dS \tag{78}$$

$$= J_{1d} \int_{S_b} \exp\left[-\frac{1}{2}(\theta - b)^\top [D_b^{-\frac{1}{2}} H_b D_b^{-\frac{1}{2}}]^{\perp e}(\theta - b)\right] dS \tag{79}$$

$$= J_{1d} \frac{(2\pi)^{\frac{n-1}{2}}}{(\prod_{i \neq e}(D_{bi}^{-1} H_{bi}))^{\frac{1}{2}}}, \tag{80}$$

where $[\cdot]^{\perp e}$ indicates the directions perpendicular to the escape direction $e$. So we have

$$\gamma = \frac{1}{2\pi} \sqrt{\frac{\det(H_a D_a^{-1})}{-\det(H_b D_b^{-1})}} |H_{be}| \exp\left(-\frac{s\Delta L}{T_a} - \frac{(1-s)\Delta L}{T_b}\right) \tag{81}$$

$T_a$ and $T_b$ are the eigenvalues of $H_a^{-1} D_a$ and $H_b^{-1} D_b$ corresponding to the escape direction. We know $D_a = \frac{\eta}{2B} H_a$ and $D_b = \frac{\eta}{2B} [H_b]^+$. As $D$ must be positive semidefinite, we re-place $H_b = U_b^\top diag(H_{b1}, \cdots, H_{b(n-1)}, H_{be}) U_b$ by its positive semidefinite analog $[H_b]^+ = U_b^\top diag(H_{b1}, \cdots, H_{b(n-1)}, |H_{be}|) U_b$. Thus, we have

$$\tau = \frac{1}{\gamma} = 2\pi \frac{1}{|H_{be}|} \exp\left[\frac{2B\Delta L}{\eta}\left(\frac{s}{H_{ae}} + \frac{(1-s)}{|H_{be}|}\right)\right]. \tag{82}$$

$\square$

### A.3 PROOF OF PROPOSITION 1

*Proof.* A stationary distribution must have a balanced probability flux between valleys. So the probability flux of each valley must be equivalent,

$$P(\theta \in V_1)\gamma_{12} = P(\theta \in V_2)\gamma_{21} \tag{83}$$

As $\tau = \gamma^{-1}$, it leads to $P(\theta \in V_v) \propto \tau_v$. We normalize the total probability to 1, then we obtain the result. $\square$

## B ASSUMPTIONS

Assumption 2 indicates that the dynamical system is in equilibrium near minima but not necessarily near saddle points. It means that $\frac{\partial P(\theta, t)}{\partial t} = -\nabla \cdot J(\theta, t) \approx 0$ holds near minima $a_1$ and $a_2$, but not necessarily holds near saddle point $b$. Quasi-Equilibrium Assumption is actually weaker but more useful than the conventional stationary assumption for deep learning (Welling & Teh, 2011; Mandt et al., 2017). Under Assumption 2, the probability density $P$ can behave like a stationary distribution only inside valleys, but density transportation through saddle points can be busy. Quasi-Equilibrium is more like: stable lakes (loss valleys) is connected by rapid Rivers (escape paths). In contrast, the stationary assumption requires strictly zero flux between lakes (loss valleys). Little knowledge about density motion can be obtained under the stationary assumption.

Low Temperature Assumption is common (Van Kampen, 1992; Zhou, 2010; Berglund, 2013; Jas-trzębski et al., 2017), and is always justified when $\frac{\eta}{B}$ is small. Under Assumption 3, the probability densities will concentrate around minima and MPPs. Numerically, the 6-sigma rule may often provide good approximation for a Gaussian distribution. Assumption 3 will make the second order Taylor approximation, Assumption 1, even more reasonable in SGD diffusion.

Here, we try to provide a more intuitive explanation about Low Temperature Assumption in the domain of deep learning. Without loss of generality, we discuss it in one-dimensional dynamics. The temperature can be interpreted as a real number $D$. In SGD, we have the temperature as $D = \frac{\eta}{2B} H$. In statistical physics, if $\frac{\Delta L}{D}$ is large, then we call it Low Temperature Approximation. Note that $\frac{\Delta L}{D}$ appears insides an exponential function in the theoretical analysis. People usually believe that, numerically, $\frac{\Delta L}{D} > 6$ can make a good approximation, for a similar reason of the 6-sigma rule in statistics. In the final training phase of deep networks, a common setting is $\eta = 0.01$ and $B = 128$. Thus, we may safely apply Assumption 3 to the loss valleys which satisfy the very mild condition $\frac{\Delta L}{H} > 2.3 \times 10^{-4}$. Empirically, the condition $\frac{\Delta L}{H} > 2.3 \times 10^{-4}$ holds well in SGD dynamics. It also suggests that, we can adjust the learning rate to let SGD search among loss valleys with certain barrier heights.

## C THE STOCHASTIC GRADIENT NOISE ANALYSIS

Figure 7 demonstrates that the SGN is also approximately Gaussian on a randomly initialized ResNet with $B = 50$ on CIFAR-10. We also note that the SGN on ResNet seems less Gaussian than SGN on

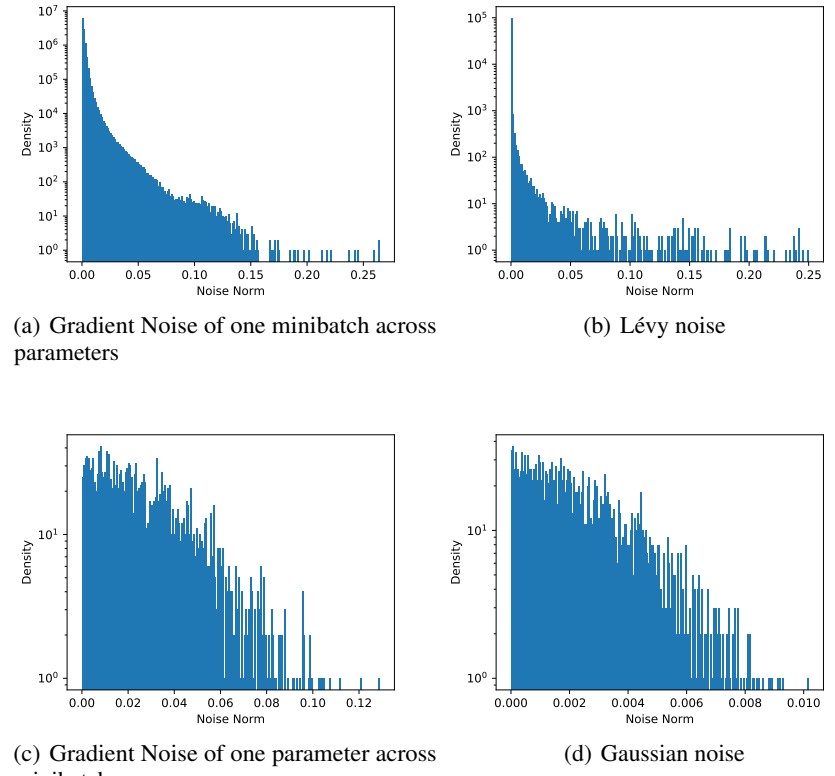

(a) Gradient Noise of one minibatch across parameters

(b) Lévy noise

(c) Gradient Noise of one parameter across minibatches

(d) Gaussian noise

Figure 7: The Gradient Noise Analysis. The histogram of the norm of the gradient noises computed with ResNet18 (He et al., 2016) on CIFAR-10 (Krizhevsky et al., 2009).

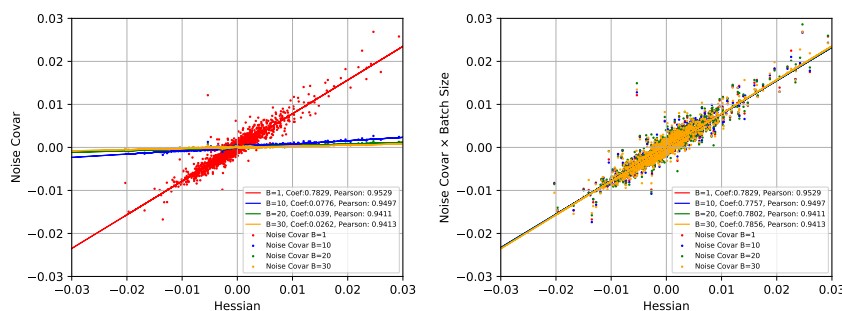

Figure 8: The plot of the SGN covariance and the Hessian by training fully-connected network on MNIST. We display all elements $H_{(i,j)} \in [-0.03, 0.03]$ of the Hessian matrix and the corresponding elements in gradient noise covariance matrix in the original coordinates.

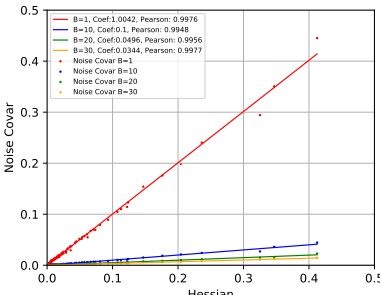 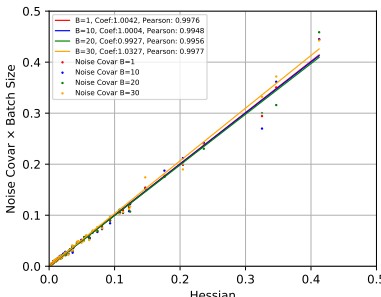

Figure 9: The plot of the SGN covariance and the Hessian by training fully-connected network on Avila. We display all elements $H_{(i,j)} \in [1e-4, 0.5]$ of the Hessian matrix and the corresponding elements in gradient noise covariance matrix in the space spanned by the eigenvectors of Hessians.

fully-connected networks with the same batch size. Panigrahi et al. (2019) presented more results on the Gaussianity of SGN under various conditions.

By Figure 8, we validate $C = \frac{H}{B}$ in the original coordinates on MNIST. By Figure 9, we also validate $C = \frac{H}{B}$ on another dataset, Avila, in the space spanned by the eigenvectors of Hessian. The relation $C = \frac{H}{B}$ can still be observed in these two cases.

**Data Precessing**: We perform the usual per-pixel zero-mean and unit-variance normalization on MNIST. We leave the preprocessing of Avila in D. **Model**: Fully-connected networks.

## D  MAIN EXPERIMENTS

Figure 10, 11, and 12 respectively validate that the exponential relation of the escape rate with the Hessian, the batch size and the learning rate.

### D.1  EXPERIMENTAL SETTINGS

**Datasets**: a) Avila, b) Banknote Authentication, c) Cardiotocography, d) Dataset for Sensorless Drive Diagnosis.

**Data Precessing**: We perform per-pixel zero-mean and unit-variance normalization on input data. For simplicity, we also transform multi-class problems into binary-class problems by grouping labels, although this is unnecessary.

**Model**: Two-layer fully-connected networks with one hidden layer and 10 neurons per hidden layer.

**Initializations**: To ensure the initialized models are near minima, we first pretrain models with 200-1000 epochs to fit each data set as well as possible. We set the pretrained models' parameters as the initialized $\theta_{t=0}$.

**Valleys' Boundary**: In principle, any small neighborhood around $\theta_{t=0}$ can be regarded as the inside of the start valleys. In our experiments, we set each dimension's distance from $\theta_{t=0}$ should be less than 0.05, namely $|\Delta\theta_i| \leq 0.05$ for each dimension $i$. If we rescale the landscape by a factor $k$, the neighborhood will also be rescaled by $k$. Although we don't know which loss valleys exist inside the neighborhood, we know the landscape of the neighborhood is invariant in each simulation.

**Hyperparameters**: In Figure 10: (a) $\eta = 0.001, B = 1$, (b) $\eta = 0.015, B = 1$, (c) $\eta = 0.005, B = 1$, (d) $\eta = 0.0005, B = 1$. In Figure 11: (a) $\eta = 0.02$, (b) $\eta = 0.6$, (c) $\eta = 0.18$, (d) $\eta = 0.01$. In Figure 12: (a) $B = 1$, (b) $B = 1$, (c) $B = 1$, (d) $B = 1$. In Figure 13: (a) $\eta = 0.0002, B = 100$, (b) $\eta = 0.001, B = 100$, (c) $\eta = 0.0002, B = 100$, (d) $\eta = 0.0001, B = 100$. In Figure 14: (a) $\eta = 0.0002, B = 100, D = 0.0002$, (b) $\eta = 0.001, B = 100, D = 0.0001$, (c) $\eta = 0.0002, B = 100, D = 0.0005$, (d) $\eta = 0.0001, B = 100, D = 0.0003$. We note that the hyperparameters need be tuned for each initialized pretrained models, due to the stochastic property of deep learning.

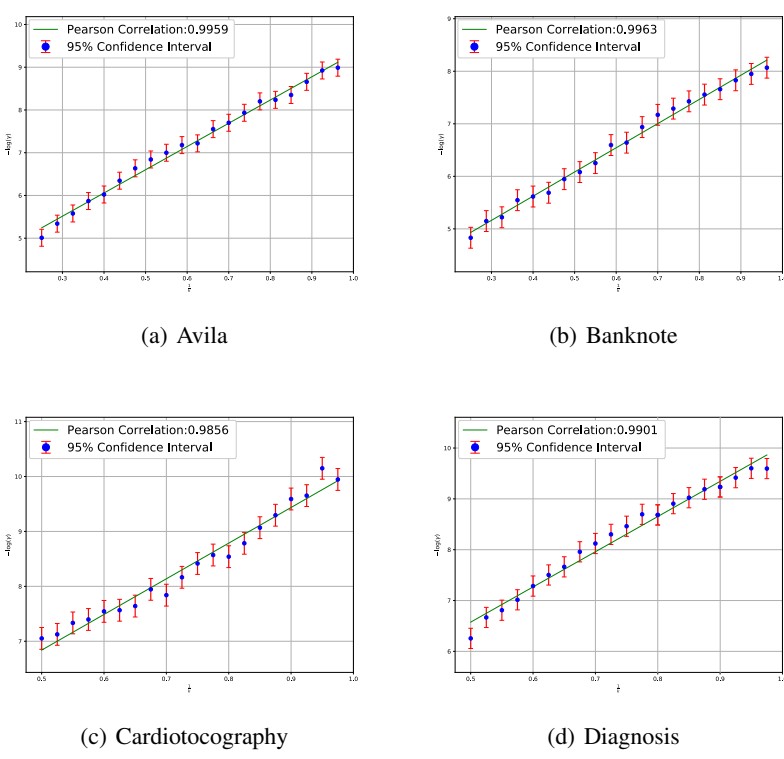

(a) Avila

(b) Banknote

(c) Cardiotocography

(d) Diagnosis

Figure 10: The escape rate exponentially depends on the "path Hessians" in the dynamics of SGD. $-\log(\gamma)$ is linear with $\frac{1}{k}$. The "path Hessians" indicates the eigenvalues of Hessians corresponding to the escape directions.

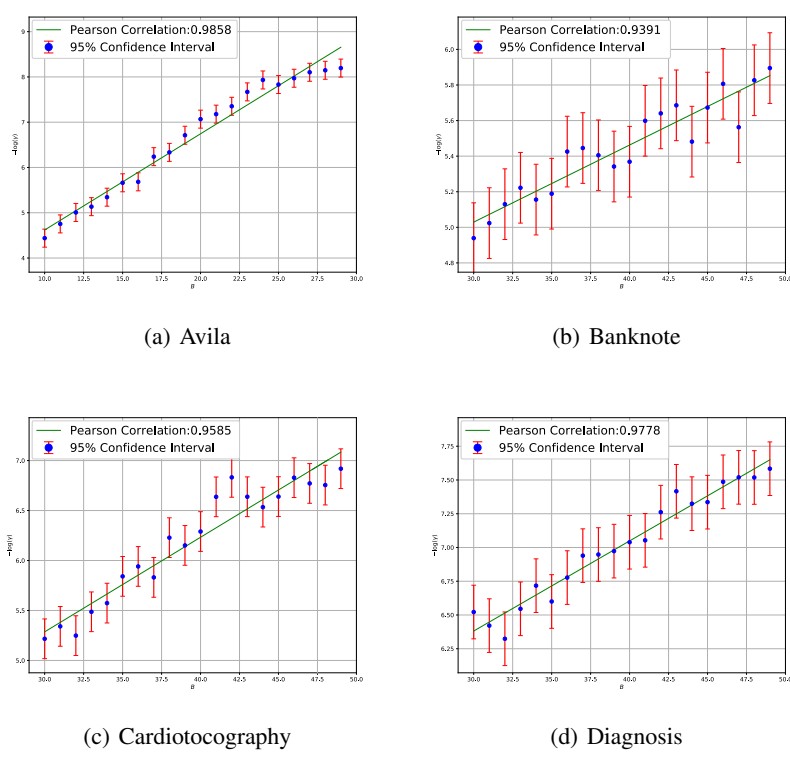

(a) Avila

(b) Banknote

(c) Cardiotocography

(d) Diagnosis

Figure 11: The escape rate exponentially depends on the batch size in the dynamics of SGD. $-\log(\gamma)$ is linear with $B$.

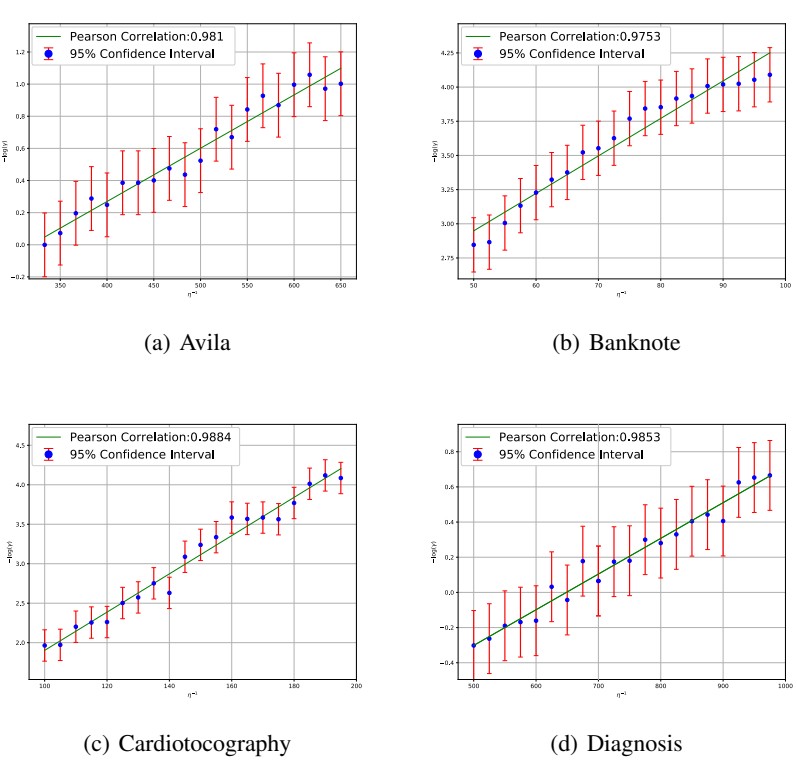

(a) Avila

(b) Banknote

(c) Cardiotocography

(d) Diagnosis

Figure 12: The escape rate exponentially depends on the learning rate in the dynamics of SGD. $-\log(\gamma)$ is linear with $\frac{1}{\eta}$. The estimated escape rate has incorporated $\eta$ as the time unit.

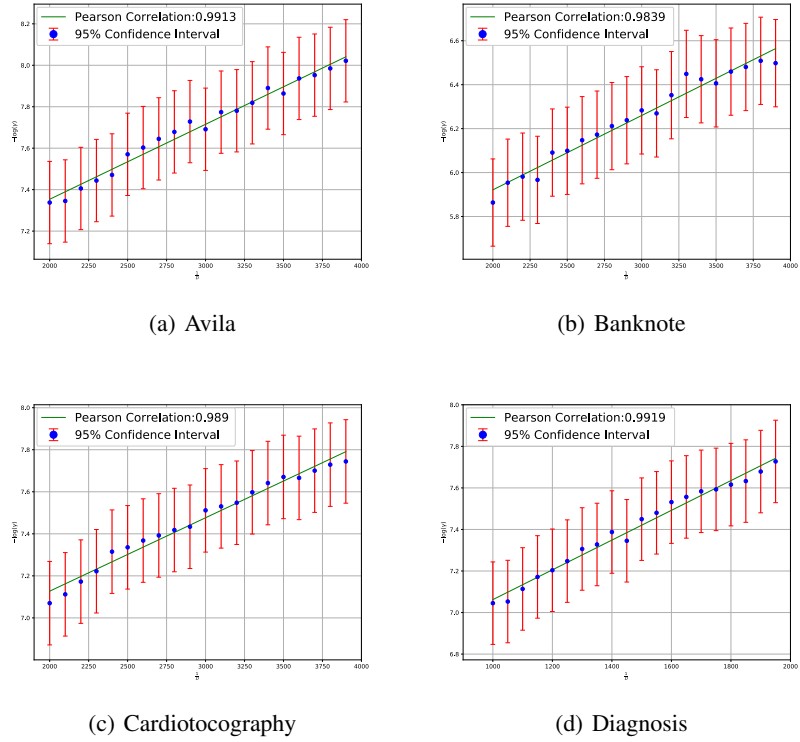

(a) Avila

(b) Banknote

(c) Cardiotocography

(d) Diagnosis

Figure 13: The relation of the escape rate and the isotropic diffusion coefficient D. The escape formula that $-\log(\gamma)$ is linear with $\frac{1}{D}$ is validated.

According to our experience, we can always find the hyperparameters to discover the quantitative relations as long as the pretrained model fits the data set well enough. The fined-tuned requirement can be avoided in Section E, because the models in Section E are artificially initialized.

**Observation**: we observe the number of iterations from the initialized position to the terminated position. We repeat experiments 100 times to estimate the escape rate $\gamma$ and the mean escape time $\tau$. As the escape time is a random variable obeying an exponential distribution, $t \sim Exponential(\gamma)$, the estimated escape rate can be written as

$$\hat{\gamma} = \frac{100 - 2}{\sum_{i=1}^{100} t_i}. \tag{84}$$

The 95% confidence interval of this estimator is

$$\hat{\gamma}(1 - \frac{1.96}{\sqrt{100}}) \leq \hat{\gamma} \leq \hat{\gamma}(1 + \frac{1.96}{\sqrt{100}}). \tag{85}$$

## D.2 EXPERIMENTS ON SGLD

**Experimental Results**: Figure 13 shows a highly precise exponential relation of the escape rate and the diffusion coefficient in the figure. Figure 14 shows a proportional relation of the escape rate and the Hessian determinant in the figure. Overall, the empirical results support the density diffusion theory in the dynamics of white noise. In experiments on SGLD, we carefully adjust the injected gradient noise scale in experiment to ensure that $D$ is significantly smaller than the loss barrier' height and large enough to dominate SGN scale. If $D$ is too large, learning dynamics will be reduced to Free Brownian Motion.

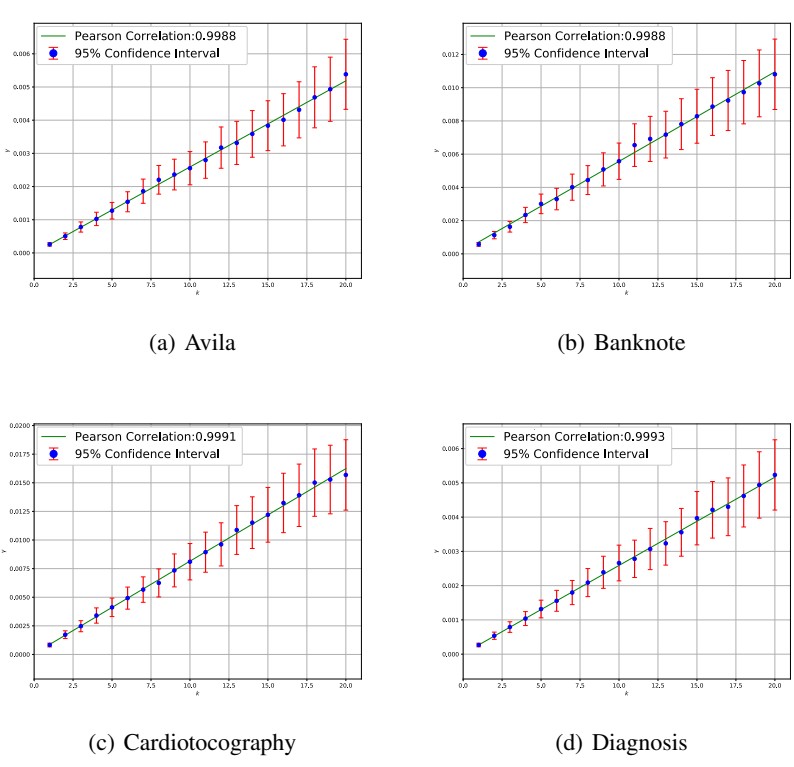

(a) Avila

(b) Banknote

(c) Cardiotocography

(d) Diagnosis

Figure 14: The relation of the escape rate and the Hessian determinant in the dynamics of white noise.The escape formula that $\gamma$ is linear with $k$ is validated.

# E   EXPERIMENTS ON MORE MODELS

We supply experiments of training three models on artificial Gaussian datasets. In these experiments, we can analytically know the locations of the minima, Hessians and loss barriers, as each input feature is Gaussian noise.

## E.1   EXPERIMENTS SETTINGS

**Data Set:** We generate 50000 Gaussian samples and random two-class labels as the training data set, $\{(x^{(i)}, y^{(i)}) | x^{(i)} \sim \mathcal{N}(0, I), y^{(i)} \in \{0, 1\}, i \in \{1, 2, \cdots, 50000\}\}$.

**Hyperparameters**: In Figure 15: (a) $\eta = 0.0001, B = 100$, (b) $\eta = 0.001, B = 100$, (c) $\eta = 0.0003, B = 100$. In Figure 16: (a) $\eta = 0.0001, B = 50, D = 0.2$, (b) $\eta = 0.001, B = 50, D = 0.0005$, (c) $\eta = 0.0003, B = 1, D = 0.0003$. In Figure 17: (a) $\eta = 0.006, B = 50$, (b) $\eta = 0.05, B = 50$, (c) $\eta = 0.005, B = 1$. In Figure 18: (a) $\eta = 0.006$, (b) $\eta = 0.06$, (c) $\eta = 0.1$. In Figure 19: (a) $B = 1$, (b) $B = 1$, (c) $B = 1$. We note that the hyperparameters are recommended and needn't be fine tuned again. The artificially initialized parameters avoids the stochastic property of the initial states.

**Experiment Setting 1:** Styblinski-Tang Function is a commonly used function in nonconvex optimization, written as

$$f(\theta) = \frac{1}{2} \sum_{i=1}^{n} (\theta_i^4 - 16\theta_i^2 + 5\theta_i).$$

We use high-dimensional Styblinski-Tang Function as the test function, and Gaussian samples as training data.

$$L(\theta) = f(\theta - x),$$

where data samples $x \sim \mathcal{N}(0, I)$. The one-dimensional Styblinski-Tang Function has one global minimum located at $a = -2.903534$, one local minimum located at $d$, and one saddle point $b = 0.156731$ as the boundary separating Valley $a_1$ and Valley $a_2$. For a n-dimensional Styblinski-Tang Function, we initialize parameters $\theta_{t=0} = \frac{1}{\sqrt{k}}(-2.903534, \cdots, -2.903534)$, and set the valley's boundary as $\theta_i < \frac{1}{\sqrt{k}}0.156731$, where $i$ is the dimension index. We record the number of iterations required to escape from the valley to the outside of valley. The setting 1 does not need labels.

**Experiment Setting 2:** We study the learning dynamics of Logistic Regression. Parameters Initialization: $\theta_{t=0} = (0, \cdots, 0)$. Valley Boundary: $-0.1 < \theta_i < 0.1$. Due to the randomness of training data and the symmetry of dimension, the origin must be a minimum and there are a lot unknown valleys neighboring the origin valley. And we can set an arbitrary boundary surrounding the origin valley group, and study the mean escape time from the group of valleys.

**Experiment Setting 3:** We study the learning dynamics of MLP with ReLu activations, cross entropy losses, depth as 3, and hidden layers' width as 10. Parameters Initialization: $\theta_{t=0} = (0.1, \cdots, 0.1)$ with a small Gaussian noise $\epsilon = (0, 0.01I)$. Valley Boundary: $0.05 < \theta_i < 0.15$. To prevent the gradient disappearance problem of deep learning, we move the starting point from the origin. For symmetry breaking of deep learning, we add a small Gaussian noise to each parameter's initial value. Due to the complex loss landscape of deep networks, we can hardly know the exact information about valleys and cols. However, the escape formula can still approximately hold even if an arbitrary boundary surrounding an arbitrary group of valleys. We set the batch size as 1 in this setting. When the batch size is small, the gradient noise is more like a heavy-tailed noise. We can validate whether or not the propositions can hold with very-small-batch gradient noise in practice.

## E.2   EXPERIMENTS RESULTS

Figure 15 shows the relation of the escape rate and the isotropic diffusion coefficient D. Figure 16 shows the relation of the escape rate and the Hessian determinant in the dynamics of white noise. Figure 17 shows the relation of the escape rate and the second order directional derivative in the dynamics of SGD. Figure 18 shows the relation of the escape rate and the batch size in the dynamics of SGD. Figure 19 shows the relation of the escape rate and the learning rate in the dynamics of SGD.

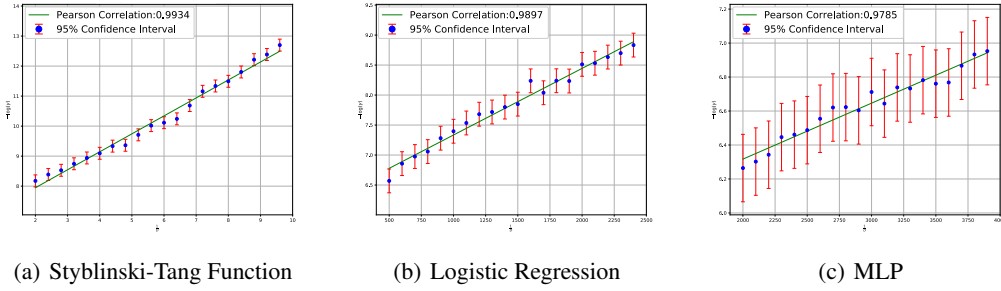

(a) Styblinski-Tang Function    (b) Logistic Regression    (c) MLP

Figure 15: The relation of the escape rate and the diffusion coefficient D in the dynamics of SGLD. The escape formula that $-\log(\gamma)$ is linear with $\frac{1}{D}$ is validated in the setting of Styblinski-Tang Function, Logistic Regression and MLP.

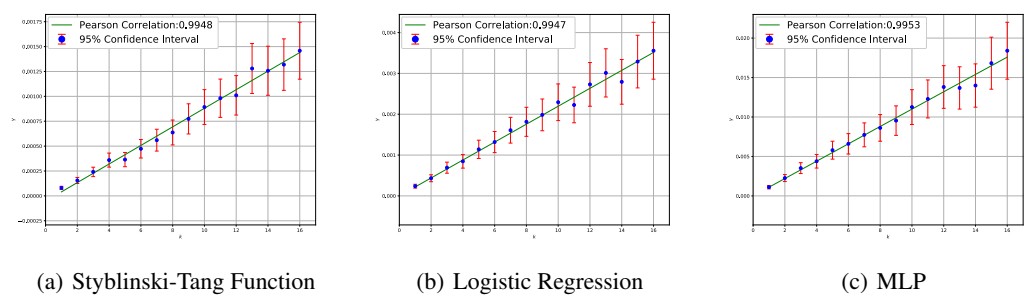

(a) Styblinski-Tang Function    (b) Logistic Regression    (c) MLP

Figure 16: The relation of the escape rate and the Hessian determinants in the dynamics of SGLD. The escape formula that $\gamma$ is linear with $k$ is validated in the setting of Styblinski-Tang Function, Logistic Regression and MLP.

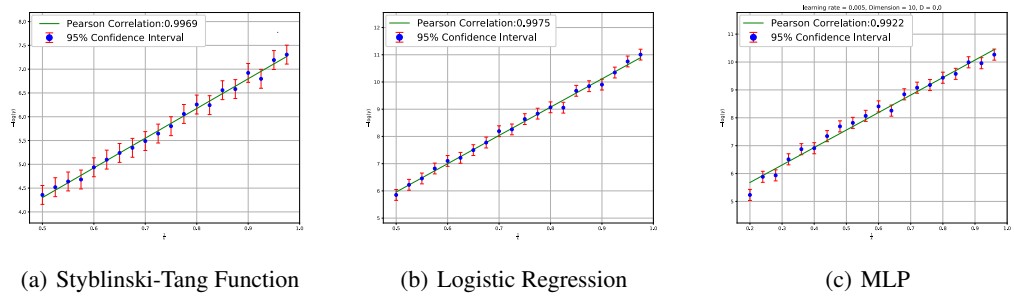

(a) Styblinski-Tang Function    (b) Logistic Regression    (c) MLP

Figure 17: The escape rate exponentially depends on the sharpness in the dynamics of SGD. The escape formula that $-\log(\gamma)$ is linear with $\frac{1}{k}$ is validated in the setting of Styblinski-Tang Function, Logistic Regression and MLP.

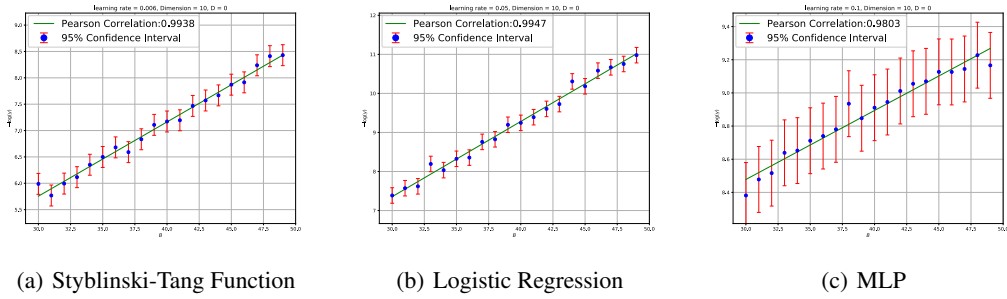

(a) Styblinski-Tang Function          (b) Logistic Regression          (c) MLP

Figure 18: The escape rate exponentially depends on the batch size in the dynamics of SGD. The escape formula that $-\log(\gamma)$ is linear with $B$ is validated in the setting of Styblinski-Tang Function, Logistic Regression and MLP.

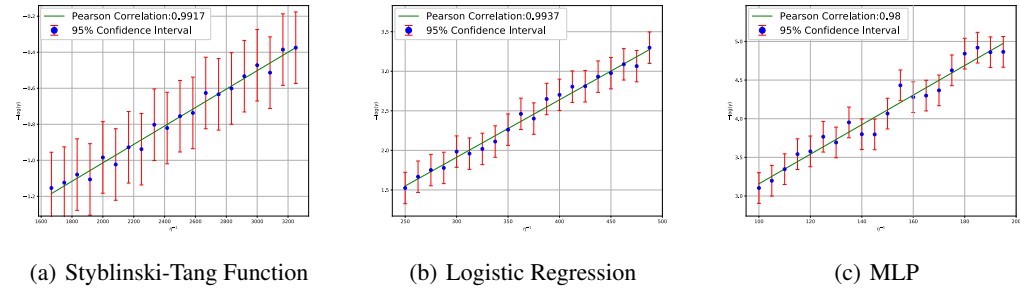

(a) Styblinski-Tang Function          (b) Logistic Regression          (c) MLP

Figure 19: The escape rate exponentially depends on the learning rate in the dynamics of SGD. The escape formula that $-\log(\gamma)$ is linear with $\frac{1}{\eta}$ is validated in the setting of Styblinski-Tang Function, Logistic Regression and MLP.

