# OpenReview forum: "A Diffusion Theory For Deep Learning Dynamics: Stochastic Gradient Descent Exponentially Favors Flat Minima"
_ICLR.cc/2021/Conference — ICLR 2021 Poster_

### Official Review · AnonReviewer4 · 2020-10-23
**Decent paper with good theory and numerics**

**Rating:** 7
**Confidence:** 4

**Review:**

Overview: The paper aims to explain how SGD converges to flatter minima using density diffusion theory and shows the effects of Hessian-dependent covariance noise versus injected white noise. Clearly, the direction is very interesting and relevant and I think this is a good paper with a clear analysis and numerics which validate the theory.

Questions, comments and ideas:
- I like the fact that this work shows how Hessian-dependent covariance noise compares to white noise. I had seen some empirical results on this (as you also mention), so it is very nice to see some theoretical validation for this. I also think this paper improves on previous related other research (e.g. the Levy noise analysis).
- The analysis and assumptions are clear. Assumptions 1-3 are of course a bit restrictive but as you also mention classical and needed to facilitate the analysis.
- What are the implications of Assumption 1? How does it depend on the network architecture?
- How does this work relate to the idea that Hessians can be rescaled to create arbitrarily sharp/flat minima? More generally, is flatness still the right concept to describe generalizability of the model?
- The numerical results are for a two-layer network? What happens if we were to increase depth or width?
- Perhaps I missed this but how are the multiple possible escape paths accounted for in the numerical results? Also what is the effect of having multiple escape paths and how does this depend on the model?
- In your opinion, do the results hold across different loss functions? In my own experience I have observed differences in MSE loss and classification loss.

---

> ### Author Response · Authors · 2020-11-17
> **Response To Reviewer4**
>
> We appreciate the reviewer for the hard work and helpful comments.
>
> The main concerns have been duly addressed below.
>
>
>
> Q1: What are the implications of Assumption 1? How does it depend on the network architecture?
>
> A1: Assumption 1 is a very common assumption in related work. It suggests that we are more interested in learning behaviors near critical points, as behaviors near critical points dominate minima selection. For a more intuitional understanding, we provide two key insights behind this counter-intuitive property.
> (1)  The region near the minimum $a$ has the most probability densities (due to the 6-sigma rule) and decides the stationary distribution under low temperature. The escape path between $a$ and $b$ will not affect the stationary distribution near $a$. Thus, the Hessian along the escape path does not appear in the expression of the probability “volume”, $P(\theta \in V_{a})$.
> (2)  The region near the saddle point $b$ decides the probability flux (based on the analysis of the Smoluchowski equation). The escape path between $a$ and $b$ will not affect the probability flux near $b$. Thus, the Hessian along the escape path does not appear in the expression of the probability flux.
> In summary, the region near $a$ controls the probability “volume” and the region near $b$ controls the probability flux. Thus, the mean escape time approximately only depends on the behaviors near $a$ and $b$.
>
>
>
> Q2: How does this work relate to the idea that Hessians can be rescaled to create arbitrarily sharp/flat minima? More generally, is flatness still the right concept to describe generalizability of the model?
>
> A2: Thanks for the interesting question. We think that flatness is still one of most important concepts to describe generalizability, when we properly use it. If we already know a good solution $\theta_{\star}$, we can, of course, transform it as a sharper minimum by rescaling the Hessian without harming model performance. However, if we perform the loss landscape rescaling before stochastic training, flat minima still usually generalize better. Specific manual counter-examples do not mean much. It is usually the expectation of generalization that matters to stochastic algorithms, according to the PAC-Bayesian theory.
>
>
>
> Q3: The numerical results are for a two-layer network? What happens if we were to increase depth or width?
>
> A3: In the main paper, the numerical results are for a two-layer network. We also presented the numerical results of three-layer networks and Logistic Regression in the Appendix. Increasing the depth or the width does not break the exponential relations which we theoretically proposed and numerically verified. Mathematically, the proposed theorems are independent of the dimensionality. However, increasing dimensionality may often increase the number of escape paths. We empirically observed that increasing dimensionality indeed reduces the mean escape time due to more escape paths.
>
>
>
> Q4: Perhaps I missed this but how are the multiple possible escape paths accounted for in the numerical results? Also what is the effect of having multiple escape paths and how does this depend on the model?
>
> A4: The exponential relations we empirically verified in numerical experiments are independent of the number of escape paths, so the effect of multiple escape paths has been excluded in our experiments.
> We do not know how it quantitatively depends on the model due to the complex landscape. The quantitative relation is beyond the scope of our paper. We qualitatively know that the number of escape paths positively correlates to the dimensionality of the model.
> We also note that, in the experiment of Styblinski-Tang Function, the number of escape paths is precisely equal to the dimensionality $n$ and all escape paths have the identical landscape. Our experiments show that the escape rate is indeed precisely proportional to the dimensionality, if we adjust the dimensionality of Styblinski-Tang Function from 10 to 30 or 100. This conclusion is not surprising. If there are $n$ symmetric escape paths, of course, there will be $n$ times the probability of escaping from the given valley along one path.

---

### Official Review · AnonReviewer3 · 2020-10-25
**A good submission with some clarifications needed**

**Rating:** 7
**Confidence:** 4

**Review:**

The authors analysed the behavior of stochastic gradient descent (SGD) algorithm in non-convex settings and compare the results with Langevin dynamics, namely gradient descent with additional isotropic noise.
In the paper the problem of escaping from a minimum (the old Kramers escape problem) is analysed in the light of SGD, characterizing the SGD dynamics using stochastic differential equations.

This approach has already been analysed in the literature and the authors give a good introduction to previous works.

The main result of the paper is the characterization of mean escape time from the basin for SGD (Thm.3.2) and Langevin dynamics (Thm.3.1). The result shows how SGD exploits the anisotropicity of the landscape to favor flat minima.

**Points that needs to be clarified :**

The authors comments on the effect of large(er) learning rate on the overall escape rate, but in that case the analysis that is based on continuous time equations will not generalize straightforwardly.

It is known in the literature that gradient descent evolves in a tiny subspace dominated by the largest eigenvectors. In the final remarks ("Low dimensional diffusion") it seems that this is a feature of SGD, but actually in the paper that they mention [Gur-Ari et al. 2018] the result shows that also gradient descent have the same feature.

**Additional comments :**

In the paper it seems that there is no dependence on the space dimension. Is that really so or is it hidden in some of the variables? I find surprising that probability of escaping from a high-dimensional basin is the same as in the low dimensional case.

The images needs larger fonts. It is impossible to read them in the printed paper and even in the pdf one needs to zoom very closely to the image to read them.

In Fig.5 and 6, the authors make experiments using neural networks. The details on the network are reported on the appendix but should be reported (maybe schematically) in the main paper instead. They are very relevant and the reader should not be forced to look and the end of the appendix to find them.

Typo after : "The mean escape time analysis of SGLD"

---

> ### Author Response · Authors · 2020-11-17
> **Response To Reviewer3**
>
> We appreciate the reviewer for the hard work and helpful comments.
>
> The main concerns have been duly addressed below.
>
>
>
> Q1: The authors comments on the effect of large(er) learning rate on the overall escape rate, but in that case the analysis that is based on continuous time equations will not generalize straightforwardly.
>
> A1: We respectfully note that the continuous time, also called dynamical time, is precisely proportional to the learning rate $\eta$. The effect of $\eta$ in terms of the step size and gradient noise has been included in our analysis. When we talk about the escape rate defined through dynamical time as we did in the paper, our comment on the exponential relation is correct. We will try to make our point more clear in the next version.
>
>
>
> Q2: In the final remarks ("Low dimensional diffusion"), it seems that this is a feature of SGD, but actually in the paper that they mention [Gur-Ari et al. 2018] the result shows that also gradient descent have the same feature.
>
> A2: The “gradient descent” expression in the paper [Gur-Ari et al. 2018] actually means SGD, as they mainly used SGD in their experiments. Thus, our conclusion agrees with [Gur-Ari et al. 2018]. Both our work and the reference [Gur-Ari et al. 2018] rarely touched gradient descent.
>
>
>
> Q3: In the paper it seems that there is no dependence on the space dimension. Is that really so or is it hidden in some of the variables? I find surprising that probability of escaping from a high-dimensional basin is the same as in the low dimensional case.
>
> A3: Thanks for the interesting comment. It is true that the diffusion analysis is general and the result is independent of the dimensionality. The role of the dimensionality is actually hidden in the loss landscape and the spectrum of the Hessian. From the viewpoint of the loss landscape, higher dimensionality often indicates more escape paths and loss valleys. This is one way for the dimensionality to affect minima selection in practice. From the viewpoint of the spectrum of the Hessian, the width and the depth of deep networks can affect the eigenvalues of the Hessian significantly. Note the depth may play a different role from the width. This is another way for the dimensionality to affect minima selection.
>
>
>
> Q4: The images needs larger fonts. The details on the network are reported on the appendix but should be reported in the main paper instead. Typo after : "The mean escape time analysis of SGLD".
>
> A4: Thank you very much for the helpful suggestions. We will make improvements in the next version.

---

### Official Review · AnonReviewer2 · 2020-10-28
**Interesting toy model and experiments, though theory is a bit far away from practice**

**Rating:** 6
**Confidence:** 3

**Review:**

The paper develops a density diffusion theory to reveal how minima selection quantitatively depends on the minima sharpness and the hyperparameters. It shows theoretically and empirically that SGD favors flat minima exponentially more than sharp minima. In particular, the paper analyzed the dependence of mean escape time from the valley with the Hessians on local minima and saddle points for both SGD and SGLD, and revealed the exponential dependence of the mean escape time with the sharpness. Experiments on real-world data have verified the theoretical results on the mean escape time.


1. The main contribution of the paper is the exact characterization of the dependence of mean escape time with Hessians. And the exponential dependence on the sharpness seems to be new, with good experimental validations. Could the authors compare their theoretical results with that in the asymmetric valley paper? Are we sure the eigenvalues of Hessians are the right quantities to look at here?


2. Assumption 3 appears very vague to me: what is the mathematical formulation for a small gradient noise? Is there a threshold to determine when it is small enough such that the theory can be applied? With this simplification, the authors can greatly simplify the integration using second-order Taylor expansion around the critical points. However, the conclusion only depends on the Hessian on the minima and the saddle point instead of the Hessian information along the optimization path, which is counter-intuitive to me. I would like to see some discussion on why it is the case for mean escape time.


3. Proposition 1 seems to be an interesting result in the distribution of stationary points. However, in high dimension cases when we have multiple valleys around, how do we define the saddle point exactly? Is it the closest saddle point for all neighborhoods? It would also be interesting to see experiments that validate the distribution of stationary points.

The writing a bit sloppy; in some \sum the indices below the sum and in the expression are not consistent, and there are also weird notations such as Fisher(theta).

I read the authors' feedback and appreciate the clarifications. I keep my score.

---

> ### Author Response · Authors · 2020-11-17
> **Response To Reviewer2 (Q1-Q3)**
>
> We appreciate the reviewer for the hard work and helpful comments.
>
> The main concerns have been duly addressed below.
>
>
>
> Q1: Could the authors compare their theoretical results with that in the asymmetric valley paper? Are we sure the eigenvalues of Hessians are the right quantities to look at here?
>
> A1: Thanks for the interesting question. The second order Taylor approximation (Assumption 1) is common or even necessary in current diffusion-based quantitative analysis. If we apply Assumption 1 near critical points, there will only exist symmetric valleys in diffusion analysis. Thus, the asymmetric valley is beyond the scope of our work.
>
> However, if we may formulate SGD diffusion theory under the third order Taylor approximation near critical points in future, it will be natural to include asymmetric valleys in the diffusion theoretical framework. Note that the third order effect is much smaller than the second order effect under the low temperature assumption in Kramers Escape Problems. Thus, the top eigenvalues of Hessians are still key quantities, while the third derivatives of loss, including the asymmetric quantities, may play a relatively minor role.
>
>
>
> Q2: Assumption 3 appears very vague to me: what is the mathematical formulation for a small gradient noise? Is there a threshold to determine when it is small enough such that the theory can be applied?
>
> A2: We would like to explain the mathematical formulation for a small gradient noise and when Low Temperature Approximation is numerically reasonable. Without loss of generality, we discuss it in one-dimensional dynamics. The temperature can be interpreted as a real number $D$. In SGD, we have the temperature as $D = \frac{\eta}{2B} H$. In statistical physics, if $\frac{\Delta L}{D}$ is large, then we call it Low Temperature Approximation.
>
> Note that $\frac{\Delta L}{D}$ appears insides an exponential function in the theoretical analysis. People usually believe that, numerically, $\frac{\Delta L}{D} > 6$ can make a good approximation, for a similar reason of the 6-sigma rule in statistics. In the final training phase of deep networks, a common setting is $\eta=0.01$ and $B=128$. Thus, we may safely apply Assumption 3 to the loss valleys which satisfy the very mild condition $\frac{\Delta L}{H} > 2.3 \times 10^{-4}$. Empirically, the condition $\frac{\Delta L}{H} > 2.3 \times 10^{-4}$ holds well in SGD dynamics. It also suggests that, we can adjust the learning rate to let SGD search among loss valleys with certain barrier heights we expect. We will add the discussion into our paper.
>
>
>
> Q3: The conclusion only depends on the Hessian on the minima and the saddle point instead of the Hessian information along the optimization path, which is counter-intuitive to me. I would like to see some discussion on why it is the case for mean escape time.
>
> A3: We respectfully note that the counter-intuitive property is indeed interesting and is why the original physics work on Kramers Escape has wide applications these years. Actually, our proofs can explain why the property holds. For a more intuitional understanding, we provide two key insights behind this counter-intuitive property.
> (1)  The region near the minimum $a$ has the most probability densities (due to the 6-sigma rule) and decides the stationary distribution under low temperature. The escape path between $a$ and $b$ will not affect the stationary distribution near $a$. Thus, the Hessian along the escape path does not appear in the expression of the probability “volume”, $P(\theta \in V_{a})$.
> (2)  The region near the saddle point $b$ decides the probability flux (based on the analysis of the Smoluchowski equation). The escape path between $a$ and $b$ will not affect the probability flux near $b$. Thus, the Hessian along the escape path does not appear in the expression of the probability flux.
> In summary, the region near $a$ controls the probability “volume” and the region near $b$ controls the probability flux. Thus, the mean escape time approximately only depends on the behaviors near $a$ and $b$.

---

> ### Author Response · Authors · 2020-11-17
> **Response To Reviewer2 (Q4)**
>
>
> Q4: Proposition 1 seems to be an interesting result in the distribution of stationary points. However, in high dimension cases when we have multiple valleys around, how do we define the saddle point exactly? Is it the closest saddle point for all neighborhoods? It would also be interesting to see experiments that validate the distribution of stationary points.
>
> A4:  Thanks for the interesting comment. A brief answer is that we can always reduce multiple-valley escape into multiple pairs of two-valley escape. Thus, we only need to find the saddle points that connect two valleys as we analyzed in the paper. When we try to analyze the escape rates in the case of multiple valleys, we may simply compute the escape rates connecting two valleys one after one, which is same as the two-valley case.
>
> If we try to analyze the stationary distribution in a multi-valley case, we may consider each valley as a discrete solution and the escape rate between two valleys as the transition probability per unit time between two solutions. Thus, the minima selection process can be approximately treated as a continuous-time Jump process, and the escape rates connecting two valleys are the elements in the transition probability matrix. After enough time, called “relaxation time” in statistical physics, the probability mass may converge to those “low-energy” and “flat” solutions.
>
> There are several interesting questions in this direction. Suppose we know all hyperparameters before training a deep network. Is it possible to predict how long the “relaxation time” will be before training? Is it possible to predict how good the stationary distribution will be before training? As our paper has proposed a useful theoretical tool, we think it will be possible to analyze how these answers depend on the hyperparameters in near future. We will continue to explore this promising direction and propose more powerful tools.

---

### Official Review · AnonReviewer1 · 2020-10-29
**The proof is novel, but some assumptions need further discussion.**

**Rating:** 6
**Confidence:** 3

**Review:**

This paper develops a density diffusion theory (DDT) to reveal how minima selection quantitatively depends on the minima sharpness and the hyperparameters. In particular, this paper theoretically and empirically prove that SGD favors flat minima exponentially more than sharp minima, while gradient descent (GD) with injected white noise favors flat minima only polynormially more than sharp minima.

This paper is the first to theoretically and empirically prove that SGD favors flat minima exponentially more than sharp minima, which is novel. Furthermore, the paper is well written.

Some shortcomings are listed as follows:

First, the concept “valley” is frequently used in this paper, but it seems that no formal definition has been given for “valley”.

Second, Assumption 1 assumes that the function value near $\theta^*$ can be estimated from the second order Taylor approximation. But when the Hessian matrix changes fast, the second order Taylor approximation will have large error and hence Assumption 1 will not be reasonable. Similarly, in Theorem 3.2, when the Hessian matrices $H_a$, $H_b$ and $\Delta L$ change fast, they cannot well reflect the flatness of minima.

----------------------
After rebuttal:
I thank the authors'  clarification. I keep my rating.

---

> ### Author Response · Authors · 2020-11-17
> **Response To Reviewer1**
>
> We appreciate the reviewer for the hard work and helpful comments.
>
> The main concerns have been duly addressed below.
>
>
> Q1: The concept “valley” is frequently used in this paper, but it seems that no formal definition has been given for “valley”.
>
> A1: Thanks for the comment. The concept “valley”, also called “basin” in some related papers, is a relatively common and intuitional concept. Related work usually uses this common concept directly when it is not confusing [1,2]. We provided an illustration figure to help understanding. We will try to explain it better.
>
>
> Q2: Assumption 1 assumes that the function value near $\theta_{\star}$ can be estimated from the second order Taylor approximation. But when the Hessian matrix changes fast, the second order Taylor approximation will have large error and hence Assumption 1 will not be reasonable. Similarly, in Theorem 3.2, when the Hessian matrices $H_{a}$, $H_{b}$ and $\Delta L$ change fast, they cannot well reflect the flatness of minima.
>
> A2: Assumption 1 is reasonable in the numerical experiments and is quite common in related work, discussed in “Classical Assumptions” of the main paper. We also respectfully that the third or higher order effect is much smaller than the second order effect in Kramers Escape Problems. Particularly, under the low temperature approximation, the probability densities are very close to critical points. However, we agree that our theory could be better if the third order or even higher order effect is included. The reviewer’s point is an interesting direction. We will try to further improve SGD diffusion theory by including the higher order effects in future.
>
>
> References:
>
> [1] Jastrzębski, S., Kenton, Z., Arpit, D., Ballas, N., Fischer, A., Bengio, Y., & Storkey, A. (2017). Three factors influencing minima in sgd. arXiv preprint arXiv:1711.04623.
>
> [2] Wu, L., & Ma, C. (2018). How sgd selects the global minima in over-parameterized learning: A dynamical stability perspective. Advances in Neural Information Processing Systems, 31, 8279-8288.

---

### Author Response · Authors · 2020-11-17
**Paper Revision**

We appreciate all reviewers for the hard work and helpful comments.
We also feel grateful for the reviewers' kind support to our work.

We would like to address all reviewers’ concerns in the corresponding responses.


We also updated our manuscript according to the comments.
The change we made mainly includes:
-	We presented more discussion about the low temperature assumption and when it is numerically reasonable in Appendix B.
-	We added the discussion that the dimensionality may increase the number of escape paths and change the spectrum of the Hessians in the paragraph of “multiple-path escape”.
-	We presented larger fonts for the figures.
-	We introduced the network architecture in the main paper.
-	We corrected several typos.

---

### Decision · Program_Chairs · 2021-01-07
**Final Decision**

**Decision:**

Accept (Poster)

**Comment:**

The paper analyzes the behavior of SGD using diffusion theory. They focus on the problem of escaping from a minimum (Kramers escape problem) and derive the escape time of continuous-time SGD and Langevin dynamics. The analysis is done under various assumptions which although might not always hold in practice do not seem completely unreasonable and have been used in prior work. Overall, this is a valuable contribution which is connected to some active research questions regarding the flatness of minima found by SGD (with potential connections to generalization). I would advise the authors to improve the quality of the writing and address other problems raised by the reviewers. I think this would help the paper maximize its impact.